# Activity of the mammalian DNA transposon *piggyBat* from *Myotis lucifugus* is restricted by its own transposon ends

Alison B. Hickman[1], Laurie Lannes[1,9,11], Christopher M. Furman[1,10,11], Christina Hong[1], Lidiya Franklin[1], Rodolfo Ghirlando[1], Arpita Ghosh[2], Wentian Luo[3], Parthena Konstantinidou[4], Hernán A. Lorenzi [5], Anne Grove [2], Astrid D. Haase [4], Matthew H. Wilson [3,6,7,8] & Fred Dyda [1] ✉

Members of the *piggyBac* superfamily of DNA transposons are widely distributed in host genomes ranging from insects to mammals. The human genome has retained five *piggyBac*-derived genes as domesticated elements although they are no longer mobile. Here, we have investigated the transposition properties of *piggyBat* from *Myotis lucifugus*, the only known active mammalian DNA transposon, and show that its low activity in human cells is due to subterminal inhibitory DNA sequences. Activity can be dramatically improved by their removal, suggesting the existence of a mechanism for the suppression of transposon activity. The cryo-electron microscopy structure of the *piggyBat* transposase pre-synaptic complex showed an unexpected mode of DNA binding and recognition using C-terminal domains that are topologically different from those of the *piggyBac* transposase. Here we show that structure-based rational re-engineering of the transposase through the removal of putative phosphorylation sites and a changed domain organization - in combination with truncated transposon ends - results in a transposition system that is at least 100-fold more active than wild-type *piggyBat*.

Transposable elements are mobile genetic elements that can move from location to location in the genome of their host. There are two major classes of transposable elements. Class 1 elements are retrotransposons, which use an RNA intermediate that is converted to DNA in order to be inserted into the host genome. Class 2 elements are DNA transposons that use only DNA intermediates to integrate their DNA directly. The vast majority of eukaryotic DNA transposons are of the "cut and paste" type (Fig. 1a). Transposon superfamilies have

been defined based on the shared genetic makeup and structural organization of the transposons and their encoded transposases[1,2].

In many prokaryotes, transposable elements play dynamic ecological roles due to their ability to carry exogenous genes, notably those responsible for antibiotic resistance. However, in higher organisms, the potential genotoxic effects of unregulated transposition have resulted in severe restriction of their mobility, either through mutation or by host-encoded systems that silence transposon activity[3]. For

[1]Laboratory of Molecular Biology, National Institute of Diabetes and Digestive and Kidney Diseases, National Institutes of Health, Bethesda, MD, USA. [2]Department of Biological Sciences, Louisiana State University, Baton Rouge, LA, USA. [3]Department of Medicine, Division of Nephrology and Hypertension, Vanderbilt University Medical Center, Nashville, TN, USA. [4]Laboratory of Cellular and Molecular Biology, National Institute of Diabetes and Digestive and Kidney Diseases, National Institutes of Health, Bethesda, MD, USA. [5]Laboratory of Biochemistry and Genetics, National Institute of Diabetes and Digestive and Kidney Diseases, National Institutes of Health, Bethesda, MD, USA. [6]Department of Veterans Affairs, Nashville, TN, USA. [7]Department of Pharmacology, Vanderbilt University, Nashville, TN, USA. [8]Department of Cell and Developmental Biology, Vanderbilt University, Nashville, TN, USA. [9]Present address: Structural Motility, UMR 144 CNRS/Curie Institute, PSL Research University, Paris, cedex 05, France. [10]Present address: International Flavors and Fragrances, Wilmington, DE, USA. [11]These authors contributed equally: Laurie Lannes, Christopher M. Furman. ✉e-mail: fred.dyda@nih.gov

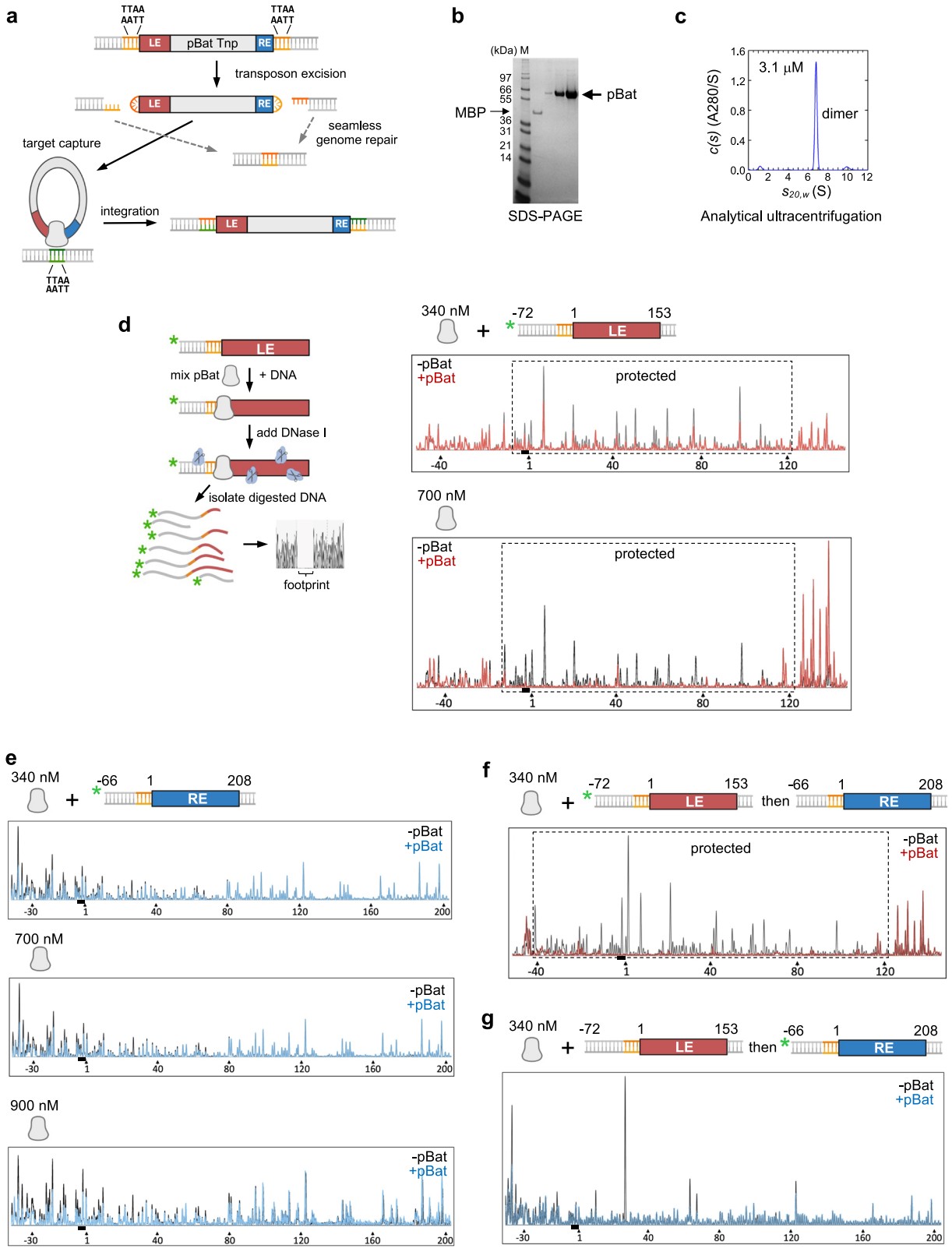

example, while about 45% of the human genome originated from transposable elements, only a small subclass of retrotransposons remains active.

Among eukaryotic DNA transposons, the eponymous member of its superfamily, *piggyBac* from the moth *Trichoplusia ni*[4], has been extensively studied and, along with *Sleeping Beauty* from the *Tc1/Mariner* superfamily[5], has become a valuable tool for genome

manipulation of human cells[6–10]. Another member of the *piggyBac* superfamily, *piggyBat*[11], is to date, the only known active DNA transposon found in mammals. It has been shown to be active in bat, yeast, and mammalian cells in culture[12,13], and it is possible that it has properties that have uniquely allowed it to remain active in a warm-blooded mammalian host, and that could be advantageous for future technological applications. On the other hand, its overall transposition

**Fig. 1 | Model of transposition and DNAse I footprinting of Left End (LE) and Right End (RE) of the *piggyBat* transposon. a** Schematic of cut-and-paste transposition by *piggyBac* superfamily members. Created in BioRender. Hickman, A. (2024) https://BioRender.com/m03j615. **b** Representative SDS-PAGE gel showing purified *piggyBat* transposase (pBat) used for footprinting and EMSA assays (purification performed >10X with similar results). M, molecular weight standards. MBP, maltose binding protein. Bands were visualized using SimplyBlue SafeStain (Novex). **c** Absorbance sedimentation c(s) profile for purified pBat transposase is consistent with a dimer. **d** Schematic of DNase I footprinting assay and footprinting profile of LE of *piggyBat*. Footprinting was carried out at room temperature by briefly incubating a fixed amount of purified DNA (~30 nM) with varying amounts of protein (30–1100 nM), followed by DNase I digestion and fragment analysis. Created in BioRender. Hickman, A. (2024) https://BioRender.com/a53z538. The green asterisk indicates the position of the fluorescent label. The black bar below the horizontal axis marks the flanking TTAA. Footprints are representative of at least three replicates. In the top trace, the red trace is with 340 nM purified pBat (shown schematically as a gray dimer), gray trace is without protein. Box with dashed outline shows protected region. In the bottom trace, the red trace is with 700 nM protein, gray trace is without protein. **e** DNase I footprinting profile of RE of *piggyBat*. Blue traces are with 340 nM protein (top), 700 nM (middle), or 900 nM (bottom) purified pBat; and matched gray traces without protein. **f** DNase I footprinting profile of LE of piggyBat in the presence of unlabeled RE. Red trace is with 340 nM protein, gray trace is without protein. **g** DNase I footprinting profile of RE of *piggyBat* in the presence of unlabeled LE. Blue trace is with 340 nM protein, gray trace is without protein. Source data are provided as a Source Data file.

activity is much lower than that of *T. ni piggyBac*, particularly when compared to *piggyBac* hyperactive variants[8,14], perhaps reflecting the current detente between transposon mobility and genome stability of its host. Neither of these aspects of *piggyBat* transposition has yet been explored.

DNA transposon ends typically have sequences organized as inverted repeats (TIRs, Terminal Inverted Repeats) that are specifically recognized by the transposase in order to synapse them and carry out DNA cleavage and joining reactions necessary to accomplish transposition. Curiously, the sequences at the *piggyBat* termini do not appear to contain the same easily recognizable pattern of short repeated subterminal motifs as observed for *piggyBac*[15,16] despite the 28.7% amino acid identity between the two transposases. As transposon end recognition is fundamental for organizing the nucleoprotein assemblies that are needed to carry out transposition in a controlled fashion (called "transpososomes"), it was unclear how the *piggyBat* transposase recognizes its ends. It is possible that DNA sequence changes accumulated through evolution that have obscured binding motifs could be responsible for the limited activity of the wild-type *piggyBat* transposon.

Here, through a combination of in vitro DNase I footprinting experiments and cell culture-based transposition assays, we have been able to rationalize the organization of the *piggyBat* transposon ends. The results allowed us to determine the cryo-EM structure of a *piggyBat* pre-synaptic complex containing one bound transposon end at 3.6 Å resolution. We have further discovered that *piggyBat*'s transposition activity in cells is restricted by an internal transposase binding site on its Left End (LE). Modifications of both the LE and the Right End (RE), elimination of predicted inhibitory N-terminal phosphorylation sites of the transposase, and tandem duplication of its C-terminal site-specific DNA binding domain following a design based on the cryo-EM structure increase transposition activity by approximately two orders of magnitude relative to wild-type, comparable to the most highly active reported *piggyBac* version[17].

## Results

### DNase I footprinting of *piggyBat* transposon ends
The *piggyBat* element in *M. lucifugus* has a single open reading frame (ORF) encoding a 572 amino acid transposase that is flanked by 586 bp on the LE and 324 bp on the RE. It has been shown that 153 bp of the LE (LE153) and 208 bp of the RE (RE208) are sufficient for activity[12], yet transposase binding motifs beyond the originally designated 15 bp TIRs[11] were not recognizable. We, therefore, expressed the transposase (pBat) in human EXPI293F cells and purified it as previously described for the *piggyBac* transposase (pB)[18] (Fig. 1b). In the absence of bound DNA, pBat was a dimer (Fig. 1c). DNase I footprinting was carried out using a purified nuclease active site mutant (pBat-D237A) to prevent unwanted DNA cleavage. A donor plasmid containing LE153/RE208 was used to generate 6-FAM labeled PCR products containing either LE153 or RE208 flanked on both sides by roughly 60 bp of nonspecific DNA.

To determine where pBat binding protects DNA from digestion, electropherograms comparing digested DNA with or without protein

were superimposed (Fig. 1d–g; separated traces are shown in Supplementary Fig. 1). On the LE, a footprint was evident that protected only ~130 bp starting a few base pairs before the TTAA target site duplication (marked with a black bar) and extending approximately to bp 120 (Fig. 1d). The footprint was partial at 340 nM pBat-D237A (Fig. 1d), but by 700 nM was fully saturated (Fig. 1d) as no further protection was observed at higher protein concentration. Surprisingly, a much higher protein concentration was required for protection on the RE (Fig. 1e), and protection remained partial even at the highest concentration used (1100 nM). At 900 nM protein (Fig. 1e), some protection appeared to extend to approximately bp 80; curiously, the strongest protection was ~40 bp into flanking DNA. As this protection was consistently observed, it might be related to the binding of a DNA end.

Considering the evidence for only weak binding of pBat to its RE, we asked if the two transposon ends bound allosterically such that binding one affects the affinity for the second. Order-of-addition experiments, in which labeled LE was preincubated with 340 nM pBat and then unlabeled RE added (Fig. 1f) or unlabeled LE was first incubated with pBat followed by addition of the labeled RE (Fig. 1g), showed that labeled LE did not show a changed footprinting pattern in the presence of RE (Fig. 1f). The protection was more complete, however, suggesting that the affinity of pBat for its LE may be enhanced in the presence of RE. Conversely, protection on the RE appeared reduced in the presence of LE (compare Fig. 1e without LE and Fig. 1g with LE), most likely due to the displacement of pBat from the RE by the LE because of the different binding affinities.

### Identification of DNA sequence motifs in the *piggyBat* LE and RE
Focusing on the DNA sequence corresponding to the ~120 bp footprint on the LE extending from the transposon end, we noticed two recurring DNA motifs (Fig. 2a, in box). One motif, 5'-GCGGGA (in green, Fig. 2a), is found at bp 11–16 (designated $G1_{LE}$), 55–60 ($G2_{LE}$), and 99–104 ($G3_{LE}$). Three bp downstream of each repeating 5'-GCGGGA motif there is an imperfect palindrome (in purple). The most interior palindrome-like sequence extends from bp 107–120 ($P3_{LE}$), corresponding to the limit of the observed footprint. There are no other occurrences of these two motifs elsewhere in the full-length 586-bp LE.

The location and spacing of the two motifs closest to the LE terminus resemble those of two repeats identified on the *piggyBac* LE (Fig. 2b, in box). The cryo-electron microscopy (cryo-EM) structure of the pB dimer bound to two LE35 oligonucleotides[18] (shown schematically in Fig. 2c, pB/LE35 complex) revealed that one motif (in green) is bound predominantly through interactions with the transposase core domain whereas the palindromic sequence (in purple) is bound by two C-terminal cysteine-rich domains (CRD), contributed by each of the monomers in the dimer.

A search for DNA motifs on the RE suggested that there are two truncated 5'-GCGGGA motifs at bp 12–15 ($G1_{RE}$) and 44–48 ($G3_{RE}$) and two more in the opposite orientation (bp 22–27, $G2_{RE}$; and 63–68, $G4_{RE}$). There is no region on the RE corresponding to the imperfect palindrome found three times on the LE. Thus, the

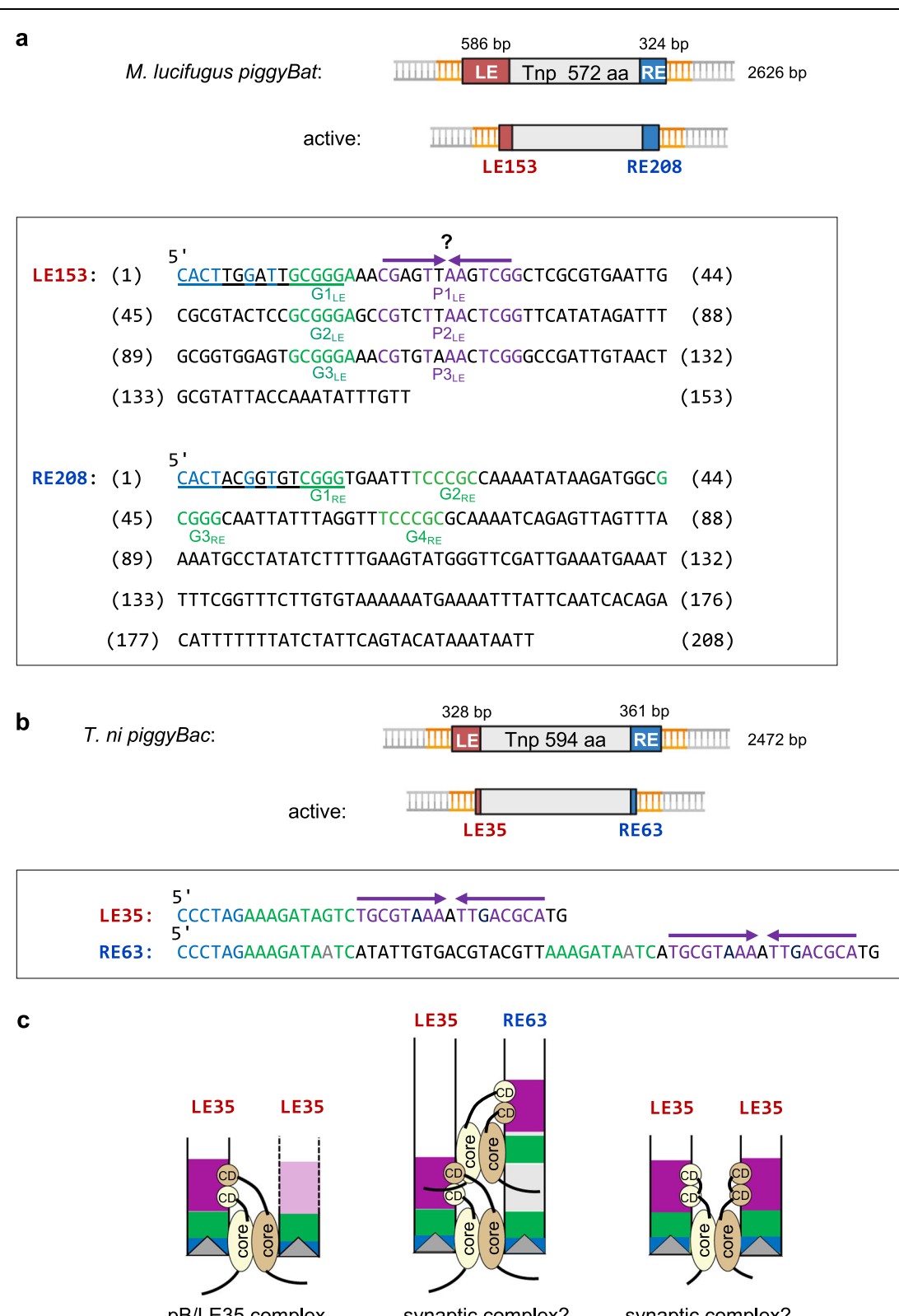

**b**

**c**

piggyBat ends differ substantially from those of piggyBac whose LE and RE contain identical 19-bp palindromic sequences arranged asymmetrically from the transposon tips (Fig. 2b). While the dearth of interpretable sequence motifs on the piggyBat RE was consistent with the weak DNase I footprint, it raised the question of how the transposase would synapse the two transposon ends, a necessity for spatially concerted integration.

## piggyBat LE shows binding patterns consistent with multiple binding sites

The extent of the DNase I footprint of the LE suggested that it contains multiple pBat binding sites and that, by analogy to piggyBac, each likely consisted of a 5′-GCGGGA motif paired with an imperfect palindrome. We performed electrophoretic mobility shift assays (EMSA) with purified pBat-D237A and labeled DNA duplexes corresponding to

**Fig. 2 | Comparison of *piggyBat* and *piggyBac* transposons. a** Schematic of the *piggyBat* transposon. Created in BioRender. Hickman, A. (2024) https://BioRender.com/m03j615. The intact transposon[11] and an active form with shorter ends comprised of 153 bp of the Left End (LE) sequence and 208 bp of the Right End (RE) sequence[12] have been described. Box: The DNA sequence corresponding to LE153 is shown on top, with repeats shown in green (designated G1$_{LE}$, G2$_{LE}$, G3$_{LE}$ from the transposon tip and purple (P1$_{LE}$, P2$_{LE}$, and P3$_{LE}$). The bases shown in purple may contribute to imperfect palindromes, as indicated by the purple arrows. The DNA sequence corresponding to RE208 is shown on the bottom, with possible repeats indicated in green. The four identical nucleotides at the transposon tips are in blue,

and the originally reported 15 bp TIRs[11] are underlined. Numbering corresponds to base pairs from each transposon end. **b** Schematic of the *piggyBac* transposon from *Trichoplusia ni* (GenBank J04364.2). Created in BioRender. Hickman, A. (2024) https://BioRender.com/m03j615. Box: Minimal transposon ends required for activity (LE35/RE63) are indicated, and repeated motifs are shown in green and purple. **c** Schematic representations of the cryo-EM structure of the pB transposase bound to two LE35 TIRs (from PDB 6x68) (left); proposed model for the pB synaptic complex (middle); and a redesigned hyperactive *piggyBac* system[17] (right). Portions adapted from ref. 17.

LE44 and LE88 of the transposon, predicted to contain one and two pBat binding sites, respectively (Fig. 3a, b). In these binding assays, when DNA at 50 nM was incubated with increasing amounts of protein (0–300 nM pBat-D237A), with LE44, we observed a single shift of the DNA band at approximately a one-to-two molar ratio of DNA-to-protein (Fig. 3a). With LE88, as the protein concentration was increased, we observed a weak band that formed at 100 nM protein, and as the protein concentration was increased, there was a second major shift accompanied by a faint ladder of higher-order complexes (Fig. 3b). None of these shifts were seen with oligonucleotide controls of unrelated sequence but of the same length. These results suggested that LE44 binds a single pBat dimer whereas LE88 binds two dimers.

To determine if the two predicted pBat binding sites in LE88 can bind pBat independently, we used oligonucleotides in which each binding site sequence was separately scrambled (Fig. 3c, d). In contrast to what was observed for LE88, when the second binding site was scrambled ("LE44scr45-88"), we observed predominantly only one shifted band, consistent with the binding of a single pBat dimer. When the first binding site was scrambled ("scr1-44LE45-88"), two shifted bands were still observed, although their relative intensity suggested that a single-bound pBat dimer was the predominant species in this case as well.

We also performed EMSA experiments using RE100. Although we did not observe a stable shift under the same assay conditions used for LE88, when we increased the concentration of RE100 to 100 nM and titrated with higher concentrations of pBat-D237A (0–600 nM), three shifted bands were detected (Fig. 3e).

As we were unable to test binding to LE132 due to the substantial difficulties we encountered correctly annealing synthetic oligonucleotides containing repeated sequences as well as presumed palindromes, we asked whether pBat interacts with the third possible binding site contained within LE89-132 (G3$_{LE}$ + P3$_{LE}$). As shown in Fig. 3f, pBat appears to independently bind all three predicted binding sites with similar affinities. We also attempted to detect synaptic complexes by EMSA in which pBat bound a LE and RE oligonucleotide simultaneously. However, we were ultimately unsuccessful, despite varying the binding conditions and order of addition (two examples are shown in Supplementary Fig. 2).

### Cryo-EM structure of the pBat pre-synaptic complex assembled on LE44

To further define the interaction between pBat and its LE, we assembled complexes of pBat bound to LE44 and solved the three-dimensional structure at 3.6 Å using single particle cryo-EM (Supplementary Fig. 3 and Supplementary Table 1). In the structure, both monomers of the dimer contribute to the binding of a single LE44 duplex (Fig. 4a). pBat possesses a large core domain consisting of the RNaseH-like catalytic subdomain that contains the DDD active site (D237, D309, D413) and a predominantly β-stranded insertion subdomain; these, in turn, are inserted into an all α-helical subdomain. There is no potential density between residues 1 and 72 suggesting disorder, although we tentatively assigned extra density observed in the target binding cleft to residues 10–27 of one of the polypeptide chains based on the AlphaFold2[19] prediction for this region. The second major domain is a cysteine-rich C-terminal site-specific DNA

binding domain (CRD) with a clear electrostatic potential density that spans residues S494 and Y572. The linker connecting the core and CRD domains is disordered in both monomers (residues 477–494 in one monomer and 480–493 in the other).

The pBat core is similar to that of pB to which it can be superimposed as a monomer between residues 87 and 475 (pB residues 119–513) at an r.m.s of 1.8 Å over 329 α-carbon positions (Fig. 4b). When the superposition is carried out using the pBat core domain colored in gold, the second core domain of pBat (i.e., that not used in the superposition, in orange) is shifted from the corresponding pB catalytic domain by as much of 13 Å when topologically identical residues of the insertion domains were compared. The paths of the pB and pBat bound donor DNA superimpose well, but the relative shift of the orange core domain means that the tip of the pBat transposon end is also about 13 Å away from the nearest DDD catalytic site (Fig. 4c) and 35 Å distant from the other active site of the dimer, a clearly catalytically-incompetent configuration. In this position, the transposon tip is stabilized by an interaction with a β hairpin (amino acids 382–399) just downstream from the insertion domain that is 11 amino acids longer in pBat than the corresponding structural element in pB (amino acids 426–433; Fig. 4d). These observations suggest that upon binding the RE to form the synaptic complex, a conformational change involving a large domain movement will occur to bring the tip of the LE to the active site. The configuration seen in the current pre-synaptic complex, if extended by flanking DNA, would prevent DNA cleavage, ensuring that pBat will not generate a double-strand break before synapse formation.

The pBat CRD is not structurally homologous to that of pB: it has a different topological fold (Fig. 5a) and recognizes a palindrome unrelated in sequence to that of pB. The pBat CRD has a small, central three-stranded antiparallel β-sheet and binds two $Zn^{2+}$ ions in a cross-brace mode, one using $Cys_3His_1$ ligands and the other with a $Cys_2His_2$ coordination sphere (Supplementary Fig. 4a). The two CRDs of the dimer together bind the imperfect palindrome (P1$_{LE}$, Fig. 2a). The symmetry axis of the CRD dimer formed on P1$_{LE}$ differs from that of the core domain, yielding an asymmetric assembly as previously observed in pB transposomes[18].

The $Zn^{2+}$ finger topology of the pBat CRD is clearly unusual as a DALI[20] search for structural homologs yielded only three similar $Zn^{2+}$ binding domains (PDB codes: 1X4S, 7SEK, and 3CXL), all contained within larger proteins with diverse functions apparently unrelated to DNA binding (Supplementary Fig. 5). In the ZNHIT2 protein, the homologous domain (1X4S) plays a role in assembly of the U5 small nuclear ribonucleoprotein particle[21,22] yet with no evidence for a role in DNA binding[21]. Methionine aminopeptidase 1 (7SEK) is involved in regulating zinc homeostasis[23] and human chimerin 1 (3CXL) is a GTPase activating protein[24]. Albeit not detected by DALI as a pBat CRD homolog, the C1 domain of protein kinase C[25], which binds the second messenger diacylglycerol, has the same topology as seen in chimerin (PDB code 7L92; Fig. 5a). Despite their topological similarity to the pBat CRD, the chimerin and PKC C1 domains have two $Cys_3His_1$ $Zn^{2+}$ binding sites. Although not yet structurally characterized, the domesticated *piggyBac*-like proteins PGBD2, PGBD3, and PGDB4 most likely have the same CRD fold as pBat based on the similarity in zinc-ligand identity and spacing in addition to AlphaFold2 prediction[19,26]

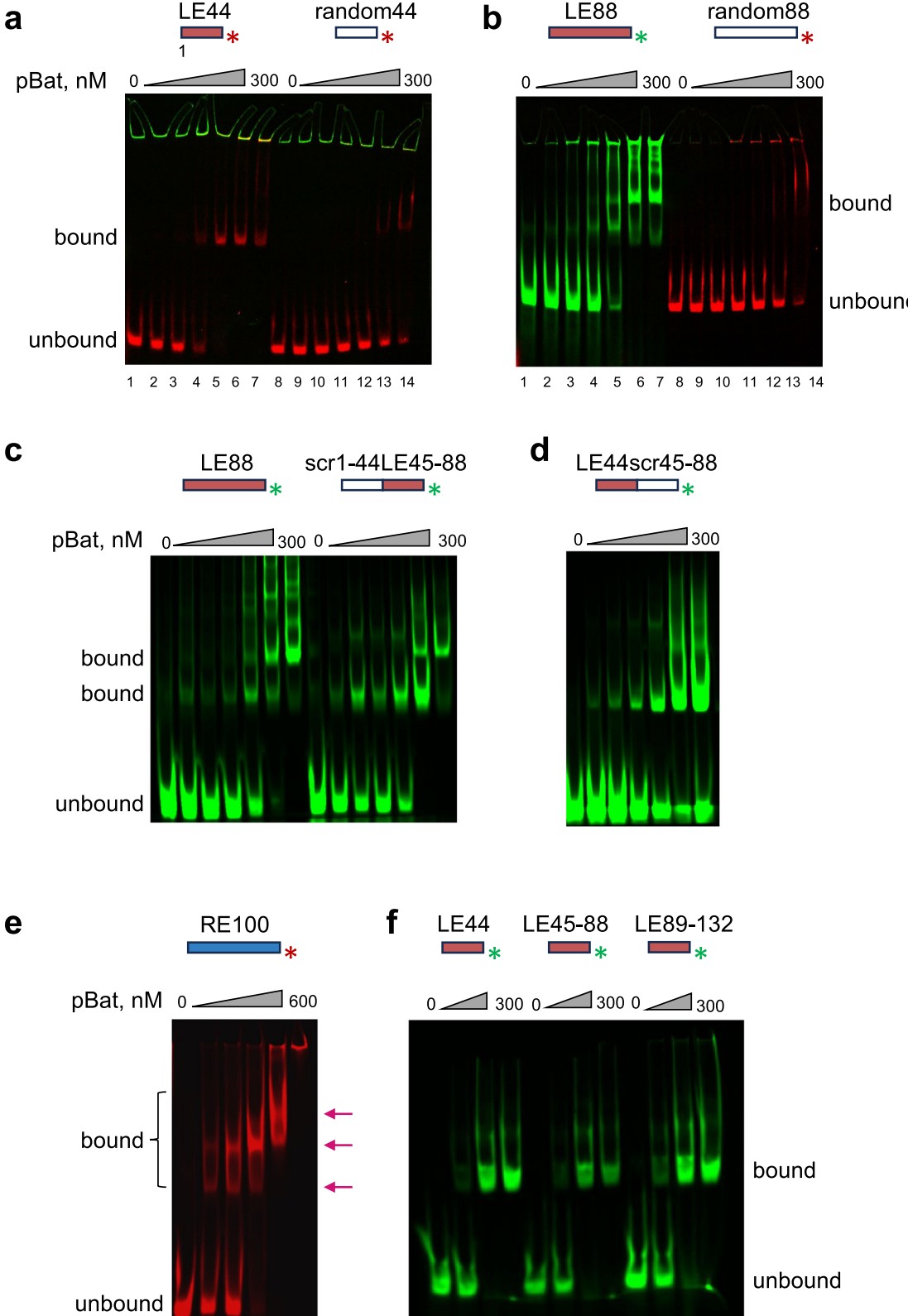

**Fig. 3 | Electromobility Shift Assays (EMSA) with *piggyBat* transposon end sequences. a** pBat transposase shows multiple interactions in an EMSA DNA binding assay. LE44 with increasing concentrations of pBat-D237A. A single shifted species was observed with LE44 (lanes 1–7; 1: 0 nM protein; 2: 12.5 nM; 3: 25 nM; 4: 50 nM; 5: 100 nM; 6: 200 nM; 7: 300 nM) but not with a random oligonucleotide of the same length (lanes 8–14; protein concentrations are the same as lanes 1–7). **b** LE88 with increasing concentrations of pBat-D237A. In this case, two major shifted species were observed with LE88 (lanes 1–7; protein concentrations as above); these were not observed with a random oligonucleotide of the same length (lanes 8–14; protein concentrations as above). The red and green asterisks indicate the color of the fluorescent label. **c** The effect on pBat-LE88 binding by scrambling ("scr") the sequence of either the first 44 bp of LE88 ("scr1-44LE45-88") or (**d**) bp 45–88 ("LE44scr45-88"). **e** pBat binding to 100 nM RE100. Left to right, lanes correspond to 0 nM protein, 50, 100, 200, 400, and 600 nM. Three shifted species were observed (indicated by arrows). **f** pBat binds independently to oligonucleotides containing the three repeated pairs of motifs on the Left End (LE44, LE45-88, and LE89-132). All experiments were performed at least two times with similar results. Source data are provided as a Source Data file.

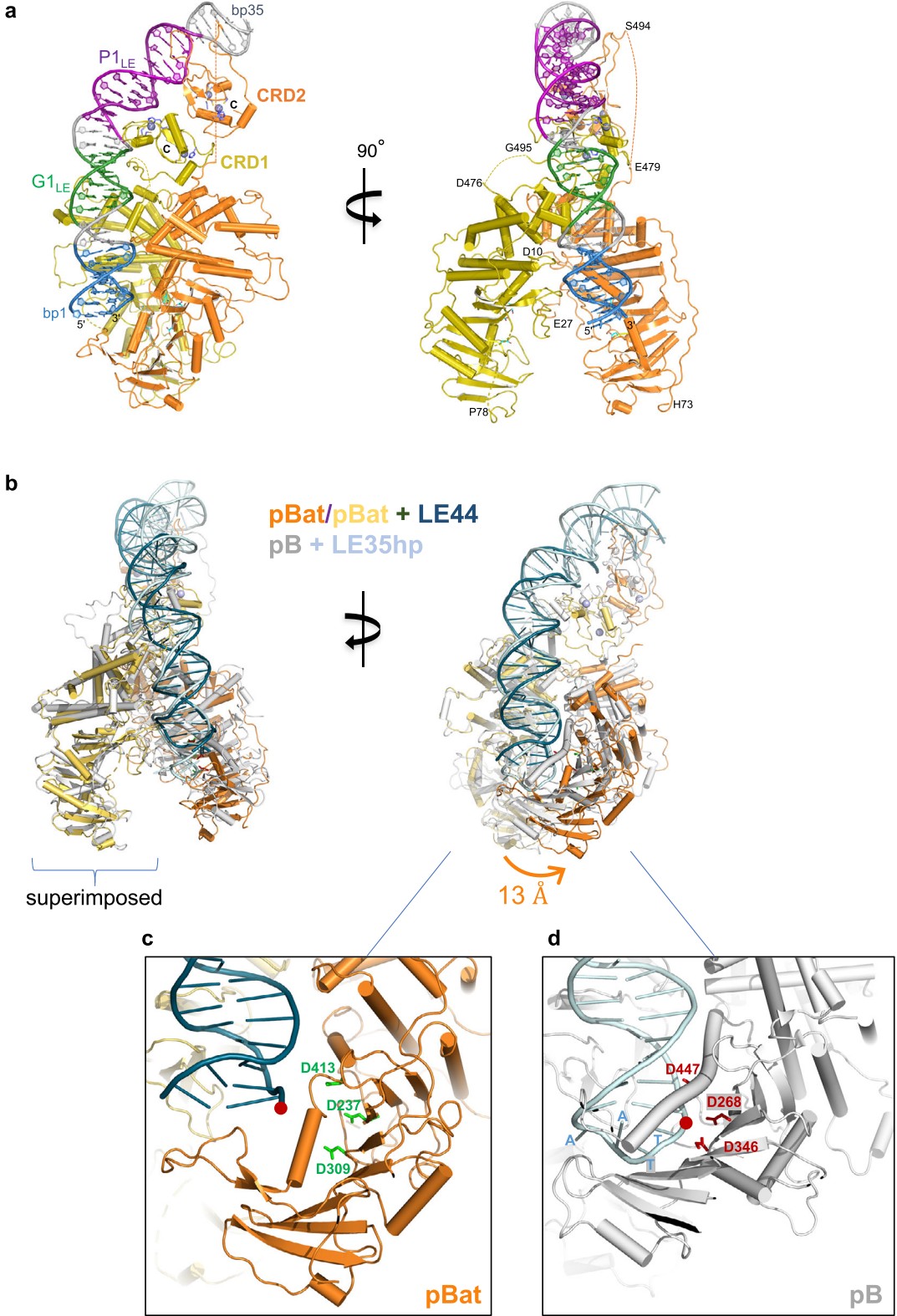

**Fig. 4 | Cryo-EM structure of pBat transposase bound to LE44. a** The two monomers of the pBat dimer are shown in gold and orange. The motifs on LE44 are colored blue (bp 1–7), green (bp 11–16), and purple (bp 19–30) as in Fig. 2a; bp 8–10, 17–18, and 31–35 are in gray. Zn²⁺ ions are shown as purple spheres, and their ligands are shown as purple sticks. Active site residues D237, D309, and D413 are shown as cyan sticks. CRD, cysteine-rich domain. **b** Structural comparison of pBat and pB bound to DNA. The coloring of pBat is as in (**a**), bound to *piggyBat* LE44, shown in dark blue; pB is shown in gray bound to a *piggyBac* LE35 hairpin (LE35hp), shown in light blue (PDB 6x68). **c** Close-up of transposon tips and the DDD catalytic triad for pBat bound to LE44; (**d**) pB bound to LE35hp. The red dots indicate the 3′-OH of LE44 and the phosphodiester bond that is broken upon hairpin opening by pB. The four bases of the TTAA hairpin are indicated for *piggyBac* LE35hp.

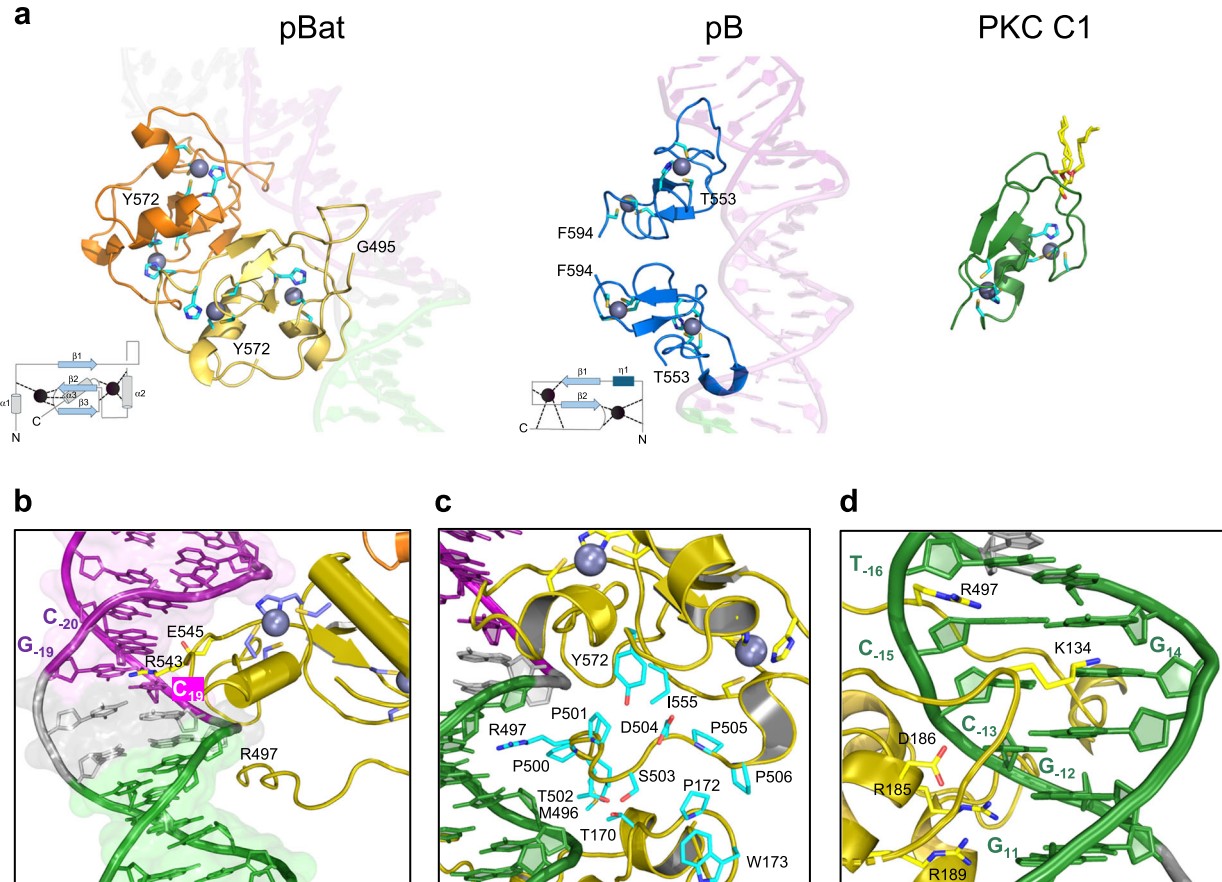

**Fig. 5 | Structural features of the pBat-LE44 complex. a** Comparison of the structures of the C-terminal domains (CRD) of pBat and pB (PDB 6x68) bound to DNA, and the protein kinase C (PKC) C1 domain (PDB 7L92). CRD topological folds are shown schematically for pBat and pB where arrows represent β-strands and cylinders are α-helices. $Zn^{2+}$ ions are shown as dark spheres. N and C denote the CRD termini. **b** Close-up of CRD1 binding to DNA. **c** Close-up of minor groove interactions involving pBat residues 495–506. **d** Close-up of the recognition of the GCGGGA motif (green in Fig. 2a).

(Supplementary Fig. 4b). (The other two exapted piggyBac proteins in mammalian genomes, PGBD1 and PGBD5, do not have CRDs).

The interactions between the pBat CRDs and the imperfect palindrome result in a ~ 60° bend in the DNA as it curls around the dimer formed by the CRDs. This is more severe than the ~ 40° bend observed in pB transpososomes and is aided by shape complementarity between the dimer and DNA as well as the A/T richness in the center of the palindrome. The transposase dimer DNA binding surface is compatible with the bent DNA, and this surface is stable due to the fixed three-dimensional relationship between the core domains and the CRD closest to the core, supported by protein/protein interactions between them. Palindrome recognition is mediated exclusively by the two CRDs largely through interactions involving two loops from each (interactions are summarized schematically in Supplementary Fig. 6). In the first half of the $P1_{LE}$ palindrome, the loop following the first β-strand of the central three-stranded beta-sheet is in the major groove where it forms the dominant share of palindrome recognition. (In PKC C1 domains, this loop is much shorter and forms the diacylglycerol binding site). Within this loop, E545 and R543 contact $C_{19}$, $G_{-19}$, and $C_{-20}$ (Fig. 5b). Sitting in the adjacent minor groove is the loop between G495 and T502 (Fig. 5c). This loop also mediates contacts between the core and CRD domains such that the overall effect is to pack the CRD and the core domain of the same monomer together. In pB transpososomes, there are no such loop/minor groove interactions, and in pBat, these interactions together with those of the first CRD seem to be responsible for the large observed bend of the donor DNA. The interactions by the second CRD with the second half of the palindrome are less well defined in the density. Collectively, these interactions differ from those of the pB CRD with DNA, due to the different CRD topologies and the pBat G495-T502/minor groove interactions that are not observed in the pB cryo-EM structures.

The core domain is largely responsible for the recognition of the GCGGGA motif and all six bases are recognized by amino acid side chains ($G_{11}$ by R189; $G_{12}$ by R185; $C_{13}$ by D186; $G_{14}$ by K134; and $C_{15}$ and $T_{16}$ by R497 of the CRD; Fig. 5d). Overall, the core domain that is linked to the CRD that interacts with the first half of $P1_{LE}$ forms most of the interactions with DNA; however, the trajectory of the DNA is such that the transposon end is directed towards the active site of the second monomer (orange in Fig. 4a). Thus, the core domain/transposon end interactions in the pre-synaptic complex are in trans, confirming the typically observed arrangement in DNA transpososomes. However, as both CRDs interact with the LE, the interactions are neither cis nor trans but both.

## Both LE and RE transposon ends are required for transposition and shortening either increases activity

In light of the structure of the pre-synaptic complex, the three sequential $G1_{LE}$ + $P1_{LE}$-like motifs on the LE strongly suggested that it contains three transposase dimer binding sites whereas the determinants for RE binding remained cryptic. To investigate the role of the various possible sequence motifs in transposition, we used a colony count transposition assay in human cells in which the transposase was expressed from one plasmid ("pHelper") and a second plasmid contained a puromycin expression cassette flanked by *piggyBat* transposon ends ("pDonor"; Fig. 6a). Upon co-transfection into HEK293T cells, wild-type (WT) pBat and the active LE153 and RE208 ends ("LE/RE") showed only moderate

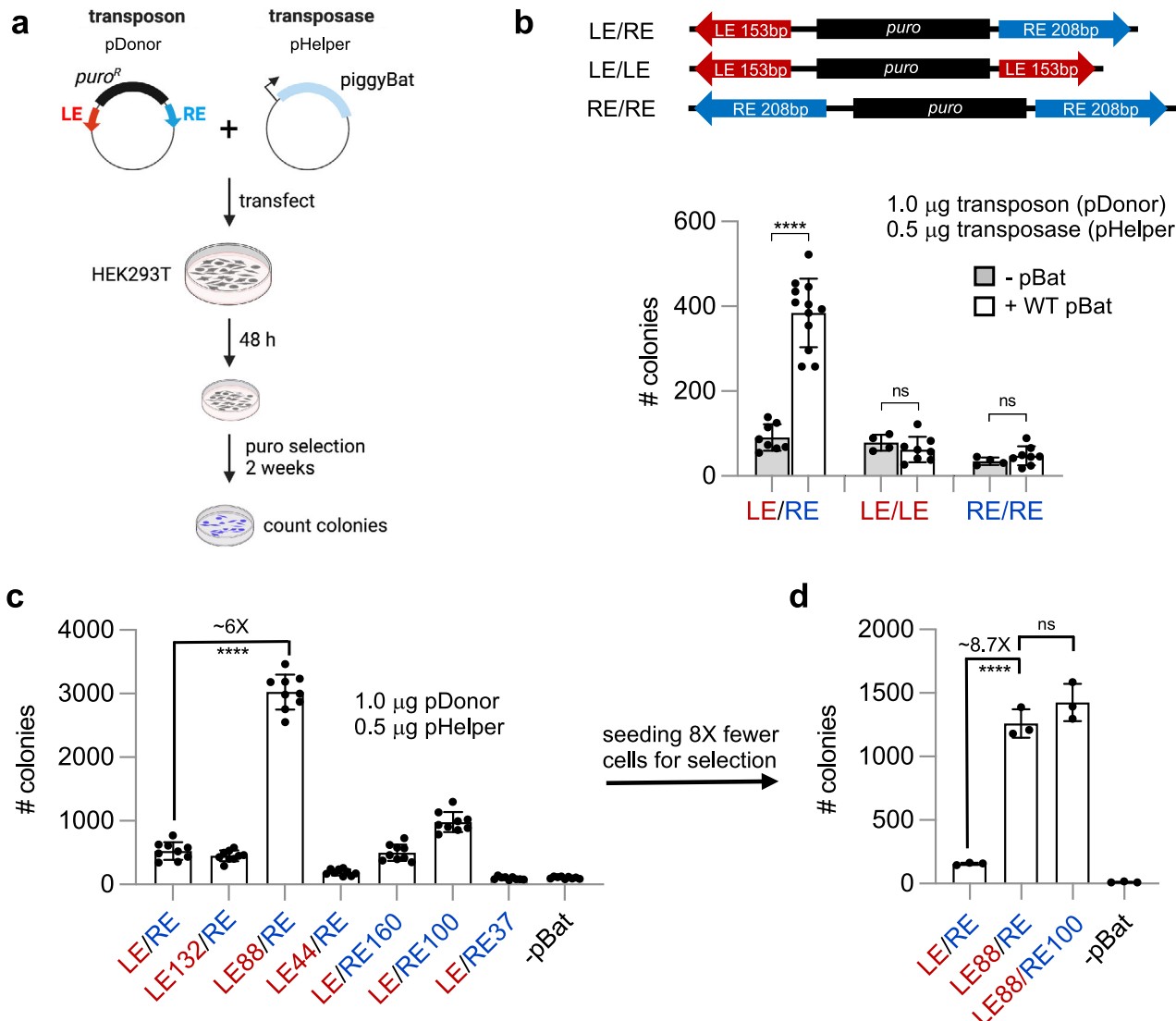

**Fig. 6 | *piggyBat* transposition in cultured human cells and the effect of truncating its transposon ends. a** Schematic of the plasmid-to-chromosome transposition assay in HEK293T cells. Created in BioRender. Hickman, A. (2024) https://BioRender.com/z01t408. **b** Transposition activity (as indicated by number of colonies) for active LE and RE of the *piggyBat* transposon with (white; $n = 12$ biological replicates) and without (gray; $n = 8$) protein in HEK293T cells. Data are presented as mean values +/− SD. No activity was detected with two LEs (LE/LE; $n = 8$ with transposase, $n = 4$ without) or two REs (RE/RE; $n = 8$ with transposase, $n = 4$ without) (as assessed by two-tailed unpaired $t$ test, GraphPad Prism). ****,

$p < 0.0001$. ns, not significant. **c** The effect on transposition activity in HEK293T cells of truncating the *piggyBat* transposon ends. LE/RE indicates LE153-RE208. 50X initial cell dilution for puromycin selection ($n = 9$ biological replicates). Data are presented as mean values +/− SD. ****, $p < 0.0001$ (two-tailed unpaired $t$ test). **d** Transposition activity with 400X initial cell dilution ($n = 3$ biological replicates). Data are presented as mean values +/− SD. The experiment has been performed twice with similar results. Each data set was compared to LE/RE using a two-tailed unpaired $t$ test. ****, $p < 0.0001$. Source data are provided as a Source Data file.

transposition (Fig. 6b) consistent with previous reports[12–14]. When "symmetrized" LE/LE or RE/RE transposon donors were used, where the donor plasmid contained identical inverted sequences, no transposition activity was observed, indicating that the asymmetric LE and RE sequences are needed together for transposition in cells.

We next investigated the effect of using shortened versions of LE and/or RE. As shown in Fig. 6c for WT pBat under our standard assay conditions (white bars), when the active RE was present, there was no change in transposition activity in truncating from the active LE (Fig. 6c; LE/RE) to LE132 (Fig. 6c; LE132/RE). However, when the innermost $G3_{LE} + P3_{LE}$ pair of motifs was deleted (Fig. 6c; LE88/RE) there was a substantial (~6-fold; $p < 0.0001$) increase in activity. Further shortening to remove $G2_{LE}$ (Fig. 6c; LE44/RE) reduced activity to only ~20% of that of the full-length WT LE/RE. A similar but less dramatic trend was observed upon truncating the RE: LE/RE160 (Fig. 6c) showed similar

activity to the active ends (LE/RE), but truncation to LE/RE100 led to a ~2-fold increase in transposition activity ($p < 0.0001$) and further truncation to LE/RE37 abolished activity. When fewer cells were used in the selection step to avoid saturating the colony count assay (Fig. 6d), truncating the LE to LE88 increased the measured activity ~8.7-fold, and combining the LE88 truncation with RE100 appeared to increase the activity slightly more, although the difference was not significant. Collectively, these data suggest that pBat activity is optimal when a transposase tetramer (dimer of dimers) assembles on LE88.

It, therefore, appears that both transposon ends contain interior (subterminal) sequences that restrict transposition activity in cells, with the most significant restriction due to the region corresponding to the $G3_{LE} + P3_{LE}$ transposase binding site on the LE. It is possible that the stimulatory effect of truncating RE160 to RE100 reveals the existence of a similar inhibitory binding site on the RE, but considerable

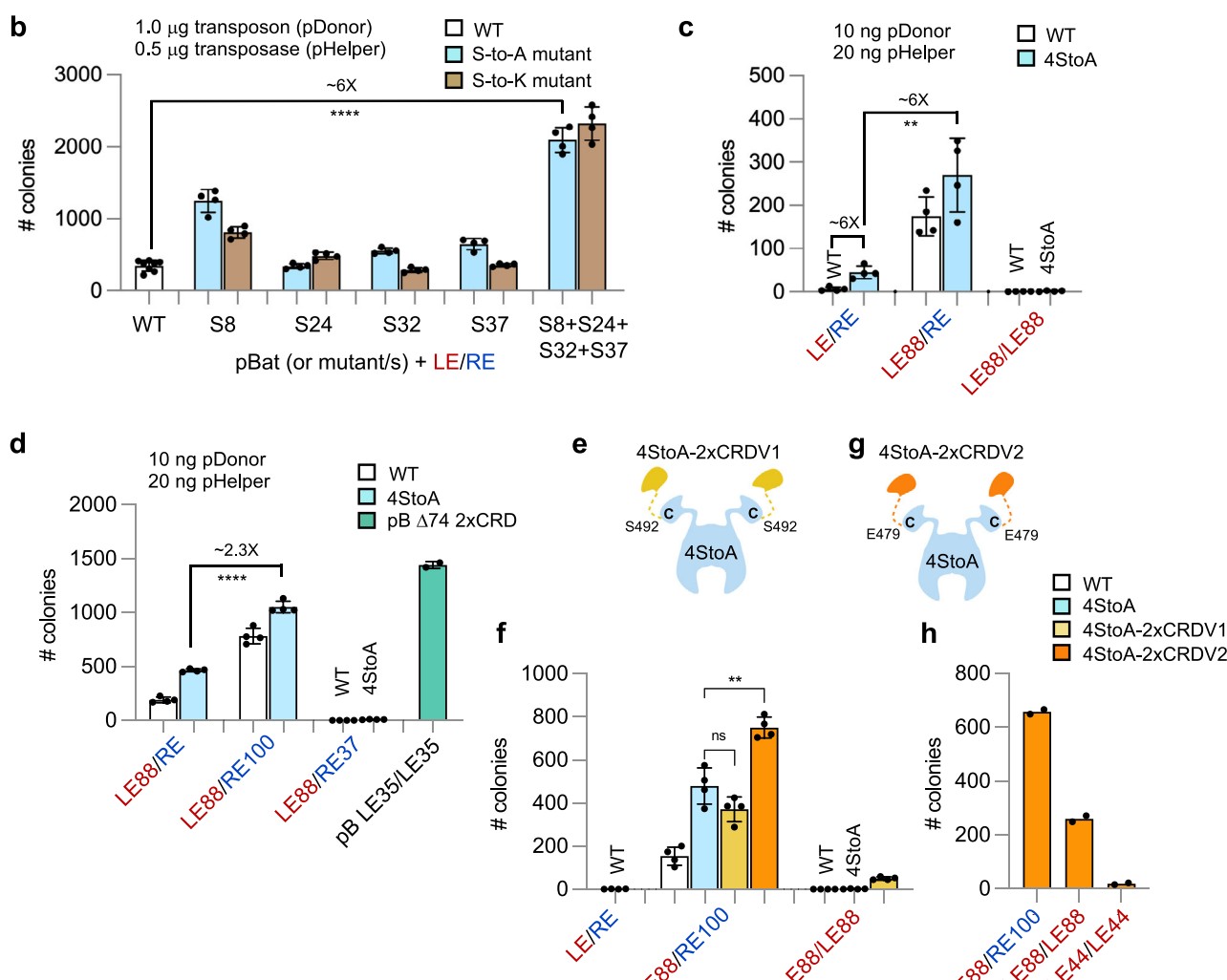

**a** CKII phosphorylation motif: **pS/pT**-D/E-X-D/E

```
              1        10        20         30          40
              .        .         .          .           .
pBat   MSQHSDYSDDEFCADKLSNYSCDSDLENASTSDEDSSDEDVMVRP
pB     MGCSLDDEHILSALLQSDDELVGEDSDSEISDHVSEDDVQSDTEE
```

**Fig. 7 | *piggyBat* transposition in cultured human cells and the effects of mutating predicted N-terminal casein kinase II phosphorylation sites and duplicating the CRD. a** pBat (top) and pB (bottom) N-terminal amino acid sequences highlighting the CKII phosphorylation motifs (underlined). The numbering above corresponds to the amino acid number of pBat. **b** Transposition activity for WT and phosphorylation mutant transposases. Data are presented as mean values +/− SD. Data for point mutants ($n = 4$ biological replicates) were compared to WT ($n = 8$) using two-tailed unpaired $t$ tests. ****, $p < 0.0001$. The experiment has been performed twice with similar results. **c** Transposition activity for WT and 4StoA point mutant transposase on truncated ends ($n = 4$ biological replicates). Data are presented as mean values +/− SD. $P$-values were determined by a two-tailed unpaired $t$ test. **, $p = 0.002$. The experiment has been performed twice with similar results. **d** Transposition activity of WT and pBat 4StoA compared to pB Δ74 2xCRD on *piggyBac* LE35/LE35. Data are presented as mean values +/− SD. $P$-values were determined by a two-tailed unpaired $t$ test. ****, $p = 0.0001$. $n = 4$ biological replicates. The experiment has been performed twice with similar results. **e** Schematic representation of pBat-4StoA-2xCRDV1. Created in BioRender. Hickman, A. (2024) https://BioRender.com/v39k454. **f** Comparison of transposition activity of pBat-4StoA, pBat-4StoA-2xCRDV1, and pBat-4StoA-2xCRDV2 on truncated *piggyBat* transposon ends ($n = 4$ biological replicates); **, $p = 0.0014$. Data are presented as mean values +/− SD. The experiment has been performed twice with similar results. **g** Schematic representation of pBat-4StoA-2xCRDV2. Created in BioRender. Hickman, A. (2024) https://BioRender.com/v39k454. **h** Transposition activity of pBat-4StoA-2xCRDV2 on truncated *piggyBat* transposon ends ($n = 2$ biological replicates). The experiment has been performed twice with similar results. Source data are provided as a Source Data file.

degeneracy must be invoked to identify a sequence that has any resemblance to the $G1_{LE} + P1_{LE}$ sequence.

## Targeted mutation of the pBat N-terminus results in hyperactivity

A defining characteristic of members of the *piggyBac* superfamily is that they contain non-conserved N-terminal regions predicted to be intrinsically disordered[15,18]. In pB, although the first 116 N-terminal amino acids showed no density in transpososome structures[18], this region contains several predicted casein kinase II (CKII) phosphorylation motifs and their removal either by N-terminal truncation or mutation results in the stimulation of transposition activity[17]. Although the N-termini of pB and pBat cannot be aligned, pBat contains four predicted CKII phosphorylation sites within its first 40 amino acids (Fig. 7a).

To determine if the predicted CKII sites impact transposition activity, we mutated the serine residues in the four CKII motifs (S8, S24, S32, S37) to alanine (A) or to lysine (K) individually or all together. As shown in Fig. 7b (in blue), three of the four alanine mutations showed some increase in activity when assayed with LE/RE, with S8A having the largest effect (~ 3-fold). When all four were mutated together ("4StoA"), the increase in activity was additive. Individual mutation of the same residues to lysine (in brown) had a statistically significant effect for S8K ($p < 0.0001$) and S24K ($p = 0.0048$), and mutation of all four to lysine led to a net ~ 6-fold stimulation of activity comparable to that of the combined serine mutations. These results suggested that phosphorylation of the N-terminus of pBat likely plays a similar role in inhibiting transposition activity as has been observed for pB.

To determine the effect of combining the 4StoA mutations with truncated transposon ends, it was necessary to adjust the assay conditions to reduce the amounts of substrates used (25X less pHelper plasmid and 100X less of pDonor). Under these conditions, the activity of WT pBat with the LE/RE donor was extremely low (in white, left bar in Fig. 7c) but allowed us to more accurately measure enhanced transposition activities of the modified *piggyBat* systems. Relative to the unmodified *piggyBat* transposon system (WT pBat), the effect of the 4StoA mutations remained on the order of an ~ 6-fold increase in activity on LE/RE, but when combined with the LE88/RE donor modification, the total measured increase was ~ 40-fold. An even greater increase relative to WT was measured when LE88/RE was replaced with LE88/RE100 (Fig. 7d).

### Duplication of the C-terminal domain allows transposition using symmetrized LE/LE ends

We recently demonstrated that the *piggyBac* transpososome can be rendered hyperactive by symmetrizing the TIRs to LE35/LE35 when we also fused an additional CRD domain to the transposase C-terminal[17] (Fig. 2c, pB Δ74-2xCRD/LE35+LE35). The design is consistent with reducing the active pB assembly from a predicted tetramer to a dimer. To determine if a similar effect might be observed for pBat, prior to the determination of the cryo-EM structure and based on sequence alignments with pB, we initially engineered a modified pBat transposase in which a second CRD was appended to the C-terminus ("pBat-2xCRDV1" comprising residues 1–572 + 492–572; Fig. 7e). However, when tested in combination with the activating 4StoA mutations ("pBat-4StoA-2XCRDV1"), there was no increase in activity on LE88/RE100 relative to pBat-4StoA; however, there was now low but detectable activity on symmetrized LE88/LE88 ends (Fig. 7f).

Once the cryo-EM structure was determined, we realized that our design could not work given the differences between the arrangement and binding mode of the pBat CRDs relative to those of pB. Specifically, we had not anticipated the ~ 43 Å distance between the C-terminus of the first CRD and the first ordered residue (S494) of the second CRD. From the cryo-EM structure of the pre-synaptic complex, the point of closest approach between the C-terminus of the first CRD and the second monomer was at residue 479, so we, therefore, generated pBat-2xCRDV2 comprising residues 1–572 + 479–572 (Fig. 7g). When this was tested in combination with the activating 4StoA mutations ("pBat-4StoA-2xCRDV2"), transposition activity on a LE88/RE100 donor was ~two-fold higher than that of pBat-4StoA (Fig. 7f). Satisfyingly, pBat-4StoA-2xCRDV2 also had significant activity on symmetrized LE88/LE88 ends (Fig. 7h) whereas pBat-4StoA was completely nonfunctional for transposition. pBat-4StoA-2xCRDV2 demonstrated only slight activity above background with a LE44/LE44 donor (Fig. 7h), despite the one-to-one correspondence between the number of CRDs and CRD binding sites. These results are in sharp contrast with our observations with pB, where the duplication of the CRDs resulted in robust activity on symmetrized LE35 donors, consistent with only a dimer of pB required to assemble on its modified transposon ends.

### Integration profiles of WT and pBat-4StoA-2xCRDV1 are indistinguishable

We determined the transposition profiles of WT pBat (with LE153/RE208) and pBat-4StoA-2xCRDV1 (with LE88/RE100) genome-wide in HCT116 cells by equipping the WT and pBat-4StoA-2xCRDV1 transposases with a self-reporting transposon cassette and identifying transcripts as surrogate for insertion sites by next-generation sequencing[27]. Analyses of sequence preferences using ggseqlogo[28] detected the characteristic transposon inverted repeat (TIR) followed by a target site duplication (TSD) for WT pBat and pBat-4StoA-2xCRDV1 (Fig. 8A). The precise excision and transposition results in strong sequence preferences surrounding the insertion site (position 1–8), while downstream sequences (positions 9–15) are highly variable representing different genomic positions and sequence content.

Next, we compared the position of WT pBat and pBat-4StoA-2xCRDV1 insertions according to their genomic annotation (Fig. 8B). More than 90% of the genome is represented by intergenic and intronic space. In contrast, protein-coding sequences (CDS) and untranslated regions (UTR) comprise less than 5%. To consider sequence requirements for transposon insertion, we normalized the genomic space according to the presence of TTAA motifs. Our analyses reveal comparable annotations for transposition events with 90% or more insertions in the intergenic and intronic space for three biological replicates and did not reveal any significant differences between WT and 4StoA-2xCRDV1 pBat insertions (Chi-square test, $p > 0.05$). However, we did observe a lower insertion frequency at intergenic regions of both constructs compared to the expected genomic representation and the genome-wide distribution of TTAA motifs that we computed based on the current genome release (GCA_000001405.15) for reference. This lower-than-expected representation of intergenic regions across all experimental conditions might be due to the reduced potential for insertions into constitutive heterochromatin, the increased fraction of repeat regions (resulting in reduced read mappability), and the larger proportion of single nucleotide polymorphisms and other sequence variants in intergenic regions compared to the reference genome.

To further characterize the distribution of insertion sites throughout the genome, we calculated the relative fraction of insertions for each chromosome (Fig. 8C). Our analysis, depicted as a heatmap in Fig. 8C showed insertions in all chromosomes for WT pBat and pBat-4StoA-2xCRDV1 transposase with notable exception of the Y chromosome, which has been lost from the HCT116 cell line[29]. Further analyses showed genome-wide distribution of new insertions for all replicates (Supplementary Fig. 7). Further inspection of a 1.4 MB region on chromosome 7 showed comparable insertion patterns by normalized read frequencies and individual insertion sites (Fig. 8D). Overall, our genome-wide analyses of transposon insertion sites showed that pBat-4StoA-2xCRDV1 maintained characteristic sequence preferences and target site duplication and is capable of transposing throughout the genome like its WT counterpart. Thus, neither the truncated transposon ends nor the introduction of the 4StoA mutations appear to have an effect on transposition fidelity.

## Discussion

*piggyBat* is the only known currently active DNA transposon found in mammalian genomes; however, its activity is very low in mammalian cells when compared to that of its close relative, *piggyBac* from *T. ni*. Low transposition activity is not itself unusual since transposons typically evolve in the absence of positive selection and they tend to accumulate debilitating mutations over time, eventually losing activity completely. For instance, *Sleeping Beauty*, a widely used DNA transposon for a variety of applications[30], originates from fossilized fragments of inactive elements in the genomes of fish species that were fused and mutated synthetically to restore transposition activity. Cells can also actively control and inhibit the activity of mobile genetic

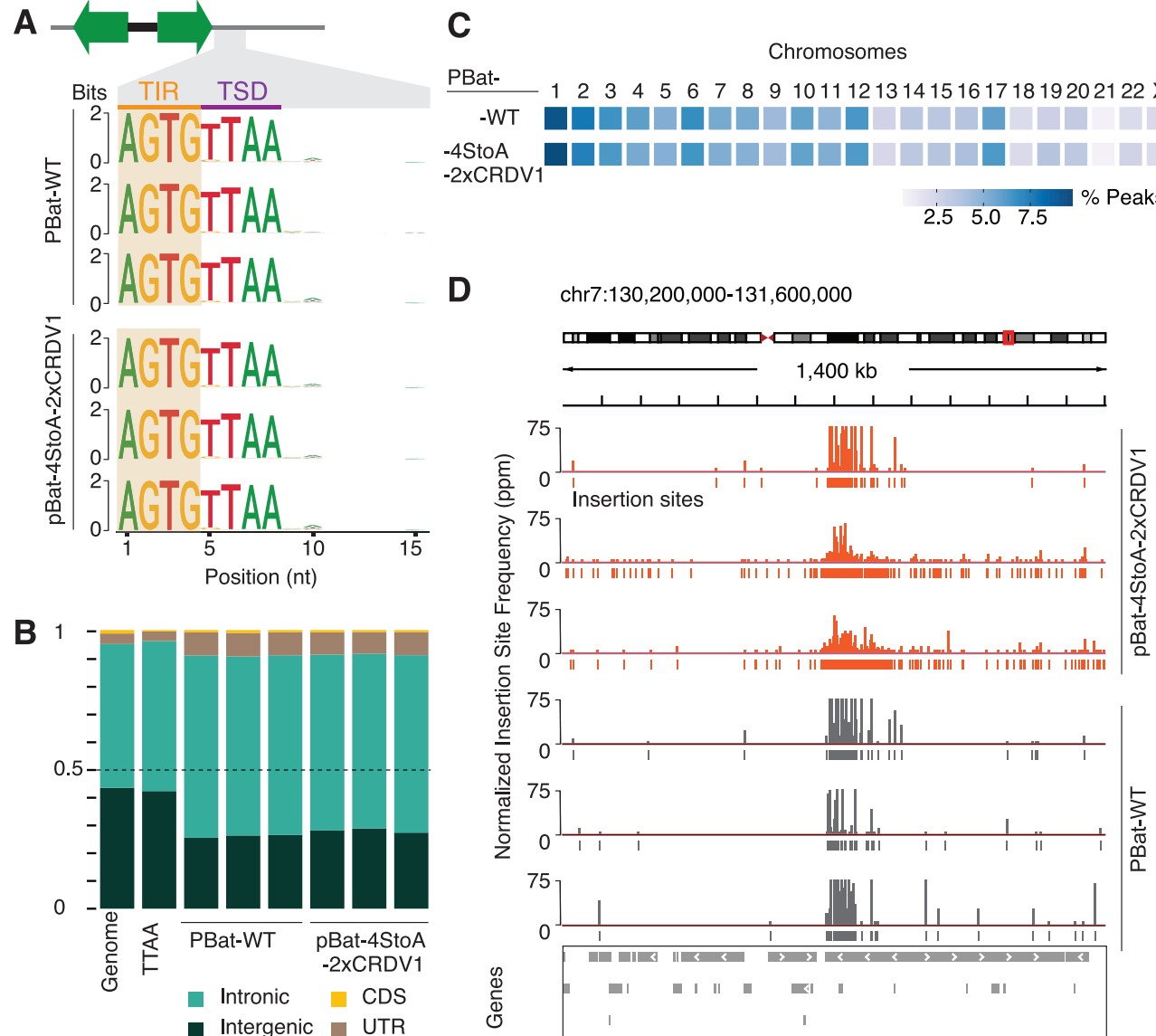

**Fig. 8 | Comparison of genome-wide insertion profiles showed comparable distribution of pBat WT and pBat-4StoA-2XCRDV1 transpositions.** Three biological replicates of independent transposition assays for WT and pBat-4StoA-2XCRDV1 in HCT116 cells were sequenced. **A** Sequence preferences of self-reported insertions by WT and pBat-4StoA-2XCRDV1 determined by Sequence Logo represent the preferred nucleotide at each position through the first 15 nucleotides (nt) including the transposon inverted repeat (TIR), followed by the duplicated target site (TSD), and the 3′ genomic region. Three biological replicates are shown for each condition. **B** Annotations of insertions by genomic features according to NCBI-RefSeq. The annotation of the entire genome is shown for comparison (first column). To account for the insertion preference into TTAA sites, we also

calculated the distribution of TTAA for each feature (second column). The biological replicates for each condition show a comparable distribution of insertion sites by genomic annotation (CDS, coding sequence; UTR, untranslated region). **C** Distribution of insertions for each chromosome. The fraction of insertions was calculated for each chromosome and depicted as a heatmap. We observed insertions throughout the genome for all replicates except for the Y chromosome, reported to be lost from HCT116 cells. **D** The density of insertions is depicted for a representative genomic region on chromosome 7. Normalized insertion frequencies are shown in peaks per million (ppm), and individual insertion sites are indicated.

elements in their genomes, for example by using the well-known piRNA pathway of transposon silencing[31]. We were interested in uncovering determinants that might restrict pBat's transposition activity in mammalian cells, given that it evolved in a mammalian background whereas pB has not.

Surprisingly, we found that a major restriction of pBat's transposition activity is due to the presence of a third transposase binding site on *piggyBat*'s LE. This is somewhat paradoxical as transposase binding to transposon ends is necessary for activity, and several transposon systems require arrays of transposase binding sites on their transposon ends to facilitate end synapse[32,33]. Among eukaryotic transposons, longer transposon sequences can be required or contribute to

increased transposition activity[34–37], an observation usually attributed to the need for repeated subterminal binding sites to enhance synapse efficiency through avidity. We are aware of only one reported example of the restriction of activity by longer, rather than shorter, terminal transposon sequences, that of the excision activity of *Stowaway* elements in the rice genome[38]. It is possible that the activity suppression by a subterminal binding site we observed here for *piggyBat* is a widespread phenomenon, as many transposon systems that have not yet been fully characterized have arrays of subterminal repeats assumed to be necessary rather than inhibitory. Our results are also a caution that although current transposition activity is one valuable metric for assessing the genome engineering potential of newly

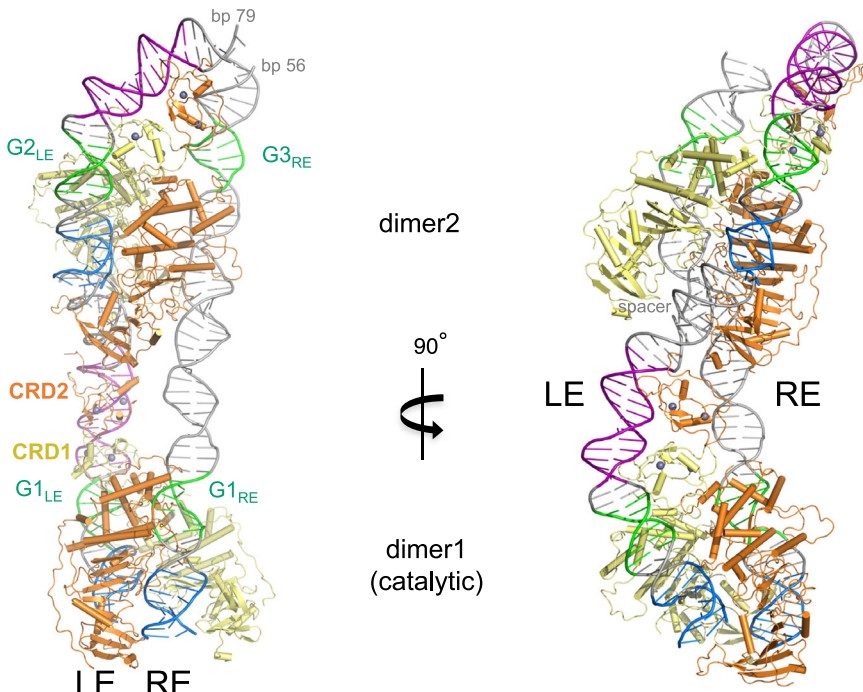

**Fig. 9 | Model of the synaptic complex of pBat bound to its transposon ends.** The model was generated using the structure of the pBat dimer bound to LE44 (where only LE bp 1–35 were visible) twice, oriented to be separated by a 9-bp B-form DNA spacer corresponding to LE bp 36–44 generated by PyMOL[73]. A model for RE bp 1–56 was also generated using PyMOL, G1_LE, and G1_RE were placed symmetrically around dimer1, and then the RE was adjusted using the Isolde feature of ChimeraX to roughly align G2_LE and G3_RE in dimer2 while avoiding protein-DNA clashes.

discovered DNA transposons, a better understanding of other features, such as the arrangement of repeated motifs within transposon ends and the consequence of their presence can be a fruitful avenue of exploration. Future studies will tell whether *piggyBat* displays genome-wide integration profiles that have been a limitation of *piggyBac* applications due to safety concerns[39]. Furthermore, it will be interesting to see whether *piggyBat* is more amenable for engineering applications aimed at specific targeting that so far have had only limited success with *piggyBac*.

While we can only speculate as to why *piggyBat's* activity is restricted by an internal transposase binding site on the LE (and possibly also by one on the RE between bp 100–160), it is likely advantageous for a transposon to suppress its mobility and hence the generation of potentially dangerous changes to its host genome. Whether an inhibitory third binding site arose from a duplication event in a transposon end that was originally more *piggyBac*-like in its requirement for a tetramer or to the degradation of an even higher order transpososome assembly is unknown. Similarly unclear is the mechanism of inhibition. Perhaps pBat binding to G3_LE prevents productive synapse of the two transposon ends by changing the optimal conformation of one of the transposon ends in the active catalytic dimer, or even causes the ends to be misaligned during end pairing. Interestingly, further shortening of the pBat LE resulted in the loss of activity suggesting that pBat requires a tetramer to assemble on its transposon ends. Tetrameric transposase assemblies are not uncommon in transpososomes[40–42], with two protomers contributing their active sites to carry out DNA cleavage and joining reactions and the other two playing an essential architectural role. WT pB, based on the distribution of transposase binding sites in its terminal repeats, must function as a tetramer. However, appending a second CRD domain to the pB transposase resulted in a highly active transposase on short symmetrized LE35/LE35 ends that likely functions as a dimer[17]. In the case of pBat with a similar approach, we measured robust activity on symmetrized LE88/LE88 ends but not with shorter LE44/LE44 ends,

indicating that pBat still requires a tetrameric assembly despite the availability of multiple CRD binding sites on both LE44 and LE88.

It would be interesting to understand what transpososome architectural necessity requires a pBat tetramer. It is possible that *piggyBat* relies on the subterminal 5′-GCGGGA motifs found on both ends to initiate the assembly of an active transpososome. The dense network of interactions between the transposase and the G1_LE motif observed in the pre-synaptic complex structure suggests a model in which one pBat dimer synapses the G1_LE/G1_RE motifs and a second dimer the G2_LE/G3_RE motifs. Using these assumptions, a model for the tetrameric assembly on LE and RE can be generated (Fig. 9) using the structure of the pre-synaptic complex: when two LE44-bound dimers are bridged with a 9-bp spacer on the LE, RE binding can be modeled with reasonable G1_LE/G1_RE and G2_LE/G3_RE juxtaposition if the DNA is allowed to bend slightly to avoid protein-DNA clashes. In such a model, there is no need for a drastic RE 60° DNA bend as observed on the LE, which is reasonable as there are presumably no CRD binding sites on the RE. We also note that such a model places CRD2 of the dimer bound at the transposon ends ("dimer1", Fig. 9) adjacent to both the β-stranded insertion domain and the first ordered N-terminal residue in the more interior dimer ("dimer2"). This suggests the possibility that tetramerization may involve the formation of an additional interface as yet unseen in any pB or pBat structures determined to date. Finally, the repeated motifs on both ends may account for the multiple bands we observed in EMSA assays carried out using only LE and RE oligonucleotides where, at the highest protein concentrations, more than two dimers appear to be bound to RE100 (Fig. 3e) and multiple complexes are formed with LE88 (Fig. 3b, c).

As robust transposition activity is a desirable property for genetic engineering applications, substantial efforts have been put into increasing activity, and the activities of some transposons can be dramatically increased by the introduction of only a small number of amino changes in the transposase and/or limited nucleotide changes in the transposon DNA (as has been done for *Sleeping Beauty*[7], Tn5[43], *piggyBac*[8], *Ac*[44]). In a sense, this mutational process might be

considered reversing evolution. We recently reported a different approach to increasing the transposition activity of *T. ni piggyBac* in which, based on three-dimensional structural information[18], the domain structure of pB was reorganized, a large deletion was applied to prevent self-inhibiting N-terminal phosphorylation, and the transposon ends were shortened, symmetrized and simplified. This strategy achieved significantly higher transposition activity than the previous point mutation-based approach. Here, the approximately two orders of magnitude activity increase for pBat by structure-based rational transposon redesign is similar to what was achieved by the large-scale mutagenic screens that resulted in SB100X[7]. The importance of structural information is further highlighted by our initial design of pBat-4StoA-2xCRDV1, which failed to account for the role of the linker between the core and CRD domains that was not predicted based on transpososome structures of *T. ni* pB.

As hinted at by sequence alignment, the DNA-binding CRD domains of pB and pBat are not structurally related. Instead, the pBat CRD domain is likely one representative of a structural family of DNA-binding domains imported into mammalian genomes[45], which includes the C-terminal domains of PGBD2, PGBD3, and PGBD4[46]. Of note, the long loop in the pBat structure between CRD strand β1 and helix α2 that is inserted into the DNA major groove (Fig. 5b) is also predicted for PGBD4 by AlphaFold2 (Supplementary Fig. 4b) and many of the DNA-contacting residues are conserved between the two (in bold, Supplementary Fig. 4a), suggesting that PGBD4 has retained its ability to bind DNA (although to date only PGBD3 has been shown to bind DNA[47]). The *pokey* transposons identified in some *Daphnia* species that target rRNA genes are also likely part of the same CRD structural group[48,49].

Clarification of the mechanism of *piggyBat* transposition - both its requirements and its restriction by additional binding sites - will have to wait for more structural information on pBat synaptic complexes. Through a combination of approaches, we have converted *piggyBat* to a transposition system that is comparable in activity to the best *piggyBac* transposition system described to date. It is our hope that *piggyBat* can now join the toolkit of highly active DNA transposons, expanding the possibilities for future genomic and clinical applications.

## Methods

### Plasmid constructs
All of the helper plasmids for expression of piggyBat transposases were based on a pFV4a-piggyBat vector derived from the pFV4a-RepHel plasmid[50] by cloning in a codon-optimized ORF of the *Myotis lucifugus* piggyBat gene using *NotI* and *SpeI* enzymes. All subsequent mutant versions were ordered as gene blocks from GenScript and cloned into the pFV4a-piggyBat plasmid. The donor plasmids which contain the TIRs were constructed from the p2NGFPmini plasmid, which contains both a puromycin resistance cassette and a GFP gene between the transposon ends, by cloning in the Left 153 bps (*NruI-XhoI*) and Right 208 bps (*NheI-SphI*) of the terminal inverted repeats of the *piggyBat* transposon. The full-length piggyBat and piggyBat-D237A transposases were codon-optimized for mammalian expression (IDT) and cloned into the pD2610 expression plasmid allowing expression of an MBP fusion protein with a TEV cleavage site[18]. All plasmid construction was performed by GenScript and confirmed by DNA sequencing.

### Purification of piggyBat and piggyBat-D237A transposases
Plasmids pD2610-pBat, or pD2610-pBat-D237A were transfected into 500 mL EXPI293F cells (ThermoFisher Scientific) for transient protein expression using PEI and harvested 3 days later and stored at − 80 °C until use. Cells expressing maltose binding protein (MBP)-tagged pBat were resuspended in lysis buffer containing 25 mM Tris-HCl pH 7.5, 500 mM NaCl, 1 mM TCEP, and protease inhibitor cocktail (Roche). The cells were lysed by sonication, and cell lysates were centrifuged at 75,000 × *g* for 45 min at 4 °C. The supernatant was filtered and mixed with 10 mL amylose resin (New England BioLabs) equilibrated with lysis buffer, and after one hour of rotation, the mixture was loaded onto a gravity flow column and washed with lysis buffer. The protein was eluted with 50 mL elution buffer (25 mM Tris-HCl pH 7.5, 500 mM NaCl, 10 mM maltose, 1 mM TCEP), and then incubated with TEV protease and dialyzed against dialysis buffer (50 mM Tris-HCl pH 7.5, 500 mM NaCl, and 1 mM TCEP) overnight at 4 °C. MBP and TEV protease were separated from pBat using a 5 mL HiTrap Heparin HP column (GE Healthcare) by linear gradient elution from 500 mM to 1 M NaCl. Purified pBat was dialyzed overnight against storage buffer (50 mM Tris-HCl pH 7.5, 500 mM NaCl, 15 % glycerol, and 1 mM TCEP), frozen, and stored at − 80 °C until use.

### Sedimentation velocity analytical ultracentrifugation (SV-AUC)
Sedimentation velocity on ~3.1 μM purified pBat in 500 mM NaCl, 25 mM Tris-HCl pH 7.4, and 0.5 mM TCEP determined its oligomeric state. Standard protocols[51] were implemented with data collected at 50,000 rpm (201,600 × *g* at 7.20 cm) and 20 °C on a Beckman Coulter ProteomeLab XL-I analytical ultracentrifuge. Sedimentation was monitored using the absorbance (280 nm) and Rayleigh interference (655 nm) optical detection systems. SEDFIT[52] was used to model the scans in terms of a continuous c(s) distribution of Lamm equation solutions, and SEDNTERP[53] provided the solution density ρ, solution viscosity η, and protein partial specific volume required for the analysis.

### DNA substrates for footprinting
DNA used in the footprinting experiments was amplified by PCR using pDonor-piggyBat-LE153/RE208 as a template to separately amplify the Left and Right End TIRs with flanking nonspecific DNA on each side of the TIR. A 247 bp region comprising the Left End (LE) was amplified using a forward primer LE-F labeled with carboxyfluorescein (6-FAM) at its 5′-end and unlabeled reverse primer LE-R. A separate 285 bp fragment comprising the Right End (RE) was amplified using unlabeled forward primer RE-F and reverse primer RE-R which was labeled with 6-FAM at its 5′ end. With the flanking DNA, this places the flanking TTAA at bp 68–71 for the Left End and bp 63–66 for the Right End. All oligonucleotide sequences are listed in the Supplementary Data file.

### DNase I footprinting
For footprinting, DNA (50 ng, corresponding to a final concentration of ~30 nM) was incubated without or with pBat-D237A protein at different concentrations for 10 min at room temperature in binding buffer (25 mM Tris-HCl pH 8.0, 100 mM NaCl, 1 mM EDTA, 0.05% (v/v) Brij58, 5 mM DTT and 2% (v/v) glycerol), following which the reaction was supplemented with 1× DNase I reaction buffer (New England BioLabs) and addition of DNase I (0.02 units). Reactions were incubated at room temperature for 3.5 min and then stopped by the addition of 8 mM Na₂EDTA. The digested DNA was purified by extraction using phenol:chloroform:isoamylalcohol (25:24:1), ethanol precipitated, dissolved in 10 μL Hi-Di formamide, and subjected to fragment analysis using an ABI 3130 analyzer after the addition of LIZ 500 ladder (ABI – Life Technologies)[54]. Microsatellite Analysis Software from ThermoFisher Cloud was used for data analysis. Electropherograms representing DNA incubated without and with protein were superimposed. In separate reactions, equimolar PCR products representing the Left End and Right End (in which only one fragment was 6-FAM-labeled) were mixed together with or without protein (340 nM or 500 nM), and samples were analyzed as indicated above.

### Cell culture, transfection, and colony count assays
HEK293T cells (obtained from and authenticated by ATCC, #CRL-11268) were cultured using standard procedures. For transfection, cells were seeded at a density of 0.5 x 10⁶ cells per well in a six-well plate and transfected one day later, initially with 1.5 μg of total plasmid DNA,

containing 1 μg of transposon (pDonor) and 0.5 μg of transposase (pHelper) plasmid DNA using Lipofectamine 3000 (Invitrogen), according to the manufacturer's instructions. For later experiments, as indicated in the figures, the amount of DNA was decreased to 10 ng pDonor and 20 ng pHelper. 48 hr post-transfection, cells were trypsinized and diluted into 100 mm dishes followed by selection with 2 μg/ml of puromycin for ~ 10 days, with media changes every three days. Either a 50-fold or 400-fold dilution was plated into selection media. Plates were then fixed using 4% formaldehyde in phosphate-buffered saline (PBS), stained with 1% methylene blue in PBS, and the colonies counted.

All statistical analyses were performed using GraphPad Prism 10.2.2. Two-tailed unpaired $t$ tests were used to determine if changes to the pBat transposon system resulted in an activity that was statistically different from the control (as indicated in each figure).

### Electrophoretic mobility Shift Assay (EMSA)
DNA binding assays were done using labeled oligonucleotides obtained from IDT. For annealing, appropriate complementary pairs of oligonucleotides (see oligonucleotide sequences in Supplementary Data file) were incubated and heated to 95 °C for 10 mins then cooled to room temperature overnight to anneal. Annealed DNA at the indicated concentrations in the figures was then incubated in buffer containing a final concentration of 25 mM Tris-HCl pH 8.0, 2 mM DTT, 5 mM MgCl$_2$, 25 mM NaCl, 1X BSA, and 8% glycerol, along with protein (pBat-D237A) at concentrations ranging from 0–300 or 600 nM for 30 min at 37 °C. Reactions were then spun down and immediately run on a 4% TBE-PAGE gel for 70 min at 120 V at 4 °C. Gels were prerun in 1XTBE for 30 min at 150 V at 4 °C to prime the gel. After running, gels were washed in water for 10 min and then imaged on a Typhoon fluorescent imager.

### Cryo-EM specimen preparation
Purified pBat transposase (5 mg/mL) and LE44 DNA were mixed in a 2:1 protein-to-DNA ratio and dialyzed overnight at 4 °C against 25 mM Tris-HCl pH 7.5, 200 mM NaCl, 0.5 mM TCEP and 5 mM CaCl$_2$. The sample was then run on a size exclusion chromatography column (Superdex 200 Increase 3.2/300) equilibrated with the sample buffer at 4 °C. The fraction containing the pBat/LE44 complex was selected moving forward. The protein concentration was estimated to be ~ 0.5 mg/mL by comparing the SDS-PAGE band intensity of the fraction with that of a dilution series of purified pBat. The pBat/LE44 complex was applied on a gold grid covered with a holey carbon film (Protochips C-Flat, R1.2/1.3, 300 mesh) freshly glow discharged for 30 s at 15 mA (PELCO easiGlow). The specimen was prepared with a Vitrobot Mark IV (FEI) rapid plunging device with the chamber at 16 °C and 100 % humidity. Three μL of sample were applied on the grid, and after 5 s the excess sample was blotted for 2.5 s (force 4) and immediately flash frozen in liquid ethane cooled by liquid nitrogen.

### Cryo-EM data collection
We collected ~ 8300 movies with a 300 kV Titan Krios TEM (FEI) equipped with a K3 direct electron detector camera (Gatan) and an energy filter (20 eV slit width). The movies were recorded in super-resolution mode at a nominal magnification of 105 kx, corresponding to a calibrated pixel size of 0.43 Å and a defocus range from − 0.8 to − 2.0 μm. The acquisition was supervised by the semi-automated program SerialEM[55]. The dose rate on the camera was set at 16.4 electrons per physical pixel per second. The total exposure time for each movie was 2.2 s with a total exposure dose of 48.8 e⁻/Å² (1.11 e⁻/Å² per frame). Each movie was composed of 44 frames (50 ms exposure per frame).

### Cryo-EM single particle analysis
The cryo-EM movies preprocessing and the single particles analysis were performed with RELION 4.0.1 run on the NIH HPC Biowulf cluster (http://hpc.nih.gov)[56–58]. The movies were motion-corrected with RELION's own implementation and binned by a factor of 2, resulting in

a pixel size of 0.86 Å for further processing. The contrast transfer function (CTF) parameters were estimated with CTFFind 4.1.14[59]. Nine hundred particles were manually picked with Topaz-Denoise turned on. The best particles from 600 movies were selected through 2D classification and used to train the neural network of the particle picker Topaz 0.2.5[60,61] to eventually automatically pick ~ 3.5 million particles on the whole dataset (0.5 picking threshold value). After extraction (240-pixel box size), the particles were purified through a first 2D classification. Two types of particles corresponding to the pBat/LE44 complex and the DNA-unbound pBat were identified and pooled into two separated selection jobs. The structure of unbound pBat was not pursued further.

### Reconstruction of the pBat/LE44 complex
The stack of pBat/LE44 particles was further purified by 2D classification/selection jobs. The resulting 795,701 particles were used to generate an initial reference-free 3D model that was used as a reference for the 3D classification of the particles (six classes). The best class (class #1) was selected (162,244 particles), and gold-standard-refined to 4.0 Å resolution at 0.143 FSC and then 3.6 Å after CTF refinement and particle polishing. The handedness of the map was corrected in ChimeraX[62,63]. The final map was sharpened and denoised with DeepEMhancer, and this map was used for model building[64].

### pBat/LE44 atomic model building
In COOT 0.9 EL[65,66], the ideal B-DNA atomic model of LE44 was generated and manually positioned into the pBat/LE44 final map. All-molecule 5 Å self-restraints were generated to perform an all-atom real-space refinement (Geman-McClure alpha set at 0.1) of the DNA duplex against the pBat/LE44 cryo-EM map. Base pairs 36 to 44, and the single-strand 5'-TTAA overhang were deleted from the model due to poor or lack of density. Two copies of the AlphaFold2[19] model of pBat were individually rigid body-fitted into the pBat/LE44 final map. The models mostly sat in the map except for the CRD domain that had to be specifically fitted (UCSF Chimera)[62]. The pBat chains were then real space refined against the cryoEM map in COOT 0.9 EL. The regions of the proteins laying outside the map were trimmed and rebuild when possible. The pBat and LE44 models were then joined into a single model that was refined using the Rosetta Relax protocol[67] followed by real-space refinement in Phenix 1.19.2.

### Preparation of libraries for next generation sequencing
HCT116 cells (obtained from and authenticated by ATCC, #CCL-247) were transfected with transposase plasmid pFV4a-pBat together with transposon plasmid pTpBat-SRT-Puro-LE-RE, and pFV4a-pBat-4StoA-2xCRDV1 with pTpBat-SRT-Puro-LE88-RE100 using lipofectamine LTX in three 100 mm dishes. The day after transfection, cells from each dish were split into four 100 mm dishes containing 3 μg/mL of puromycin medium. After two weeks of selection with puromycin, cells were collected, and RNA was prepared using the NucleoSpin RNA Plus kit (Macherey-Nagel). Four micrograms of total RNA were used to prepare cDNAs using M-MLV reverse transcriptase, RNase H minus (Promega), and primer SMART-dT18VN (see Supplementary Data file for oligonucleotide sequences). cDNAs were PCR-amplified with four primers located in transposon areas (SRT-PAC-F1, SRT-Seq P1, SRT-Seq P2, and SRT-Seq P3) and one primer (Smart) located in SMART-dT18VN. Amplified cDNAs were purified using PCR/Gel purification columns (Macherey-Nagel). PCR amplicons (500 ng) were fragmented and tagged using Illumina DNA Prep (Illumina). Tagged DNA fragments were further PCR-amplified using Read1-TnME and Read2-Pbat. PCR amplicons were 100–500 bp size-selected using a 2% agarose gel. The libraries were made twice separately and later pooled. Adapter sequences were added to these libraries and sequenced using the MiSeq or NovaSeq next-generation sequencing platform (NHLBI DNA Sequencing and Genomics Core, NIH).

## Sequencing read processing and identification of insertion sites

Sequencing reads were processed with the in-house bash script *process_forward_reads.sh*. Briefly, the PCR primer sequence spanning the 3' (LE) end of the transposon (primer_B aka Read1-TnME) and the Illumina sequencing adapters were removed from the forward reads with two consecutive runs of *cutadapt* (v4.4)[68]. Reads lacking a transposon sequence were discarded, and the remaining trimmed sequences were mapped to the human reference assembly (GCA_000001405.15) with *bowtie2* (v2.5.1). The resulting bam files were converted to bed format with the *bamtobed* tool from the *bedtools* suite (v2.31.0)[69], and the custom Perl script *bed2histogram.pl* was used to quantify the frequency of transposon insertions per site across the human genome. Identification of motifs surrounding the targeted insertion sites was carried out with the program *weblogo* (v3.7.12)[70]. Transposon insertion profiles along the human genome were visualized with the software *Integrative Genomics Viewer* (v2.16.2)[71] and *Circos*[72].

### Reporting summary

Further information on research design is available in the Nature Portfolio Reporting Summary linked to this article.

## Data availability

The atomic coordinates of the pBat/LE44 complex have been deposited in the protein data bank (PDB) with the accession code 9C0F. The EM map has been deposited in the Electron Microscopy Data Bank (EMDB) with accession code EMD-45082. Next-generation sequencing data are available at the Gene Expression Omnibus (GEO) GSE245531. Source data are provided in this paper.

## Code availability

Custom codes are available at https://doi.org/10.5281/zenodo.14248948, and may be used without restriction.

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

## Acknowledgements

Funding was provided by the Intramural Program of the National Institute of Diabetes and Digestive and Kidney Diseases of the National Institutes of Health [ZIA DK036153-16 (F.D.), 1ZIADK075111-07 (A.D.H.)], the National Institutes of Health [R01 DK093660, R01 EB033676, P30 DK114809 (M.H.W.)], and NSF award MCB-2153410 (A.Grove). The computational resources of the NIH HPC Biowulf cluster (http://hpc.nih.gov) were utilized, and next-generation sequencing was performed by the NHLBI DNA Sequencing and Genomics Core. UCSF ChimeraX was developed by the Resource for Biocomputing, Visualization, and Informatics at the University of California, San Francisco, with support from the National Institutes of Health R01-GM129325 and the Office of Cyber Infrastructure and Computational Biology, National Institute of Allergy and Infectious Diseases.

## Author contributions

C.M.F., C.H., and A.B.H. performed activity assays. C.M.F., L.F., and A.B.H. purified proteins. C.M.F. and C.H. performed the EMSAs. R.G. performed SV-AUC. A.Ghosh and A.Grove carried out the DNase I

footprinting and analysis. L.F. prepared complexes and L.L. carried out the cryo-EM experiments, determined the structure, and L.L. and F.D. analyzed the structure. W.L. and M.H.W. prepared samples for the NGS experiments, and P.K., H.L., and A.D.H. performed the subsequent analysis. A.B.H. and F.D. supervised the study. All authors participated in manuscript preparation.

## Funding

## Competing interests
This work was partially funded by a Collaborative Research and Development Agreement DK22-1043 between the NIDDK and SalioGen Therapeutics. M.H.W. previously served on the scientific advisory board of SalioGen. A.B.H., C.M.F., and F.D. have submitted a patent application (No. 63/632,275) that covers modifications to the *piggyBat* transposon system that increase its activity. The remaining authors declare no competing interests.
