## [Peer Review file · Nature Communications]

Activity of the mammalian DNA transposon piggyBat from *Myotis lucifugus* is restricted by its own transposon ends

Corresponding Author: Dr Fred Dyda

Version 0:

Reviewer comments:

Reviewer #1

(Remarks to the Author)

This manuscript from Dyda probes the transposon end binding characteristics of the piggyBat transposon, a homolog of the widely used piggybac transposon. Very detailed mechanistic studies of the binding sites and their affinities are accompanied by a partial cryo-EM structure of the transposase bound to a single end. Like piggybac, the piggybat transposase's activity is stimulated by removal of the phosphorylation site in the N-terminus. Perhaps the most interesting finding is that additional internal transposase binding sites in the piggyBat ends appear to be inhibitory to piggyBat's function.

Though the manuscript claims that the piggyBat transposon is a promising alternative to the piggyBac transposon, it isn't clear from the findings in the manuscript what is so attractive about this system compared to piggyBac. At best it appears to be a homolog that functions similarly in primary human cells. Therefore, aside from the interesting finding that internal binding sites inhibit transposition I wonder: what is the broader significance of the study aside from this? It would benefit the authors and the Nature Communications audience to have this clearly articulated.

Reviewer #2

(Remarks to the Author)

The activation of a mammalian DNA transposase and the process used to achieve it presented in this manuscript are impressive on two fronts: (1) it is likely to provide a useful new genetic tool and (2) it demonstrates the power of biochemical and structural data combined with insightful thinking when trying to optimize tools – the activation reported here would likely not have been possible even with the most powerful modern selection methods.

The work that I am most qualified to judge - the biochemical and structural sections - appears to be carefully and rigorously carried out.

Specific comments:

- 1) (very minor issue) Lines 77-80: there an abrupt transition from the idea that half the human genome derives from transposons to the next sentence that talks about DNA transposons, which (as far as we know) aren't active in the human genome.
- 2) Line 119: I see 4bp, not 15bp TIRs in Figure 1. This is confusing.
- 3) Line 132: how was "saturated" determined?
- 4) Line 166: "transposon ends are typically organized as inverted repeats" is confusing. Do you mean the TIRs (one sequence repeated once on each end)? Or the same sequence twice on each end, in inverted orientation (a palindrome)? I can only think of piggyBac as an example of the latter. In general, the discussion of motifs under the sub-heading "identification of DNA sequence motifs ..." is long-winded and confusing. The figure is nice, so a more concise summary in the text might be better.
- 5) Figure 1 – the peaks in the footprinting traces are so sharp that it is quite hard to tell red or blue from black in the traces. Can you show them offset vertically? Subtract the two curves for the main figure and put the raw curves in the supplement?
- 6) Figure 1 – how do you explain the footprint around -30 on the RE?
- 7) Having looked at the map myself in Coot, I think figure S1F slanders it. The figure made me doubt that the DNA sequence

register could be established, or that individual bases could be seen rather than a continuous stack. Looking at the map reassured me that the section on DNA sequence recognition is not over-interpreted as I'd feared. I suggest adding close-up pictures of the density supporting the protein – DNA interactions, such as those two R's diving into the major groove.

8) R497 might be contributing affinity to the protein-DNA interaction, but I don't believe it can contribute specificity because all 4 bases have H-bond acceptors at the same spot in the minor groove.

9) Can you say more about HOW the protein induces curvature in the DNA? Is it a matter of overall surface complementarity to the dimer? Asymmetric neutralization of one side of the DNA? I also noted minor groove compression in the center, where the binding site is A/T (A/T richness tends to favor narrow minor grooves – see ancient work on 434 repressor).

10) Line 369 – “figure 6D” – did you mean 7e?

11) Line 451 – “the major restriction” – because it is hard to know how many additional unknown mechanisms are tuning down transposition in vivo, this should be changed to “a major restriction”.

12) Line 506 – I see no synaptic complexes at all in figure 3C? In fact, I suspect that different fluorophores were used on the two transposon ends in order to detect synaptic complexes, and that no EMSA lanes are shown with both DNAs in the same tube because that part of the experiment failed? I would be in favor of just showing the failed experiment if that is the case – it would explain why so much effort was put into cryoEM of the single-end complex (which was, in fact, informative for protein engineering).

It may or may not be appropriate to add to this paper, but out of sheer curiosity, I would love to know how well (or poorly) the new alphafold3 server correctly docks the CRD fold onto dsDNA, since structures of that fold bound to DNA couldn't have been in its training set.

Reviewer #3

(Remarks to the Author)

Hickman et al report interesting re-engineering of the piggybac transposon. They report a novel inhibitory mechanism and they achieve remarkable activity enhancement. The authors leverage previous work from a more established piggybac system.

##general comments

-Figure 7 provides data on CRD domain. Page 13 or during the discussion (page 18) would benefit for a wider discussion on the structural insights explaining differences in V1 vs V2

-I would be great to provide more clarity on the observations from Fig. 8

1) are the differences observed in B significant?

2) it results unclear to me what is being reported in D. 3 replicated of the two different variants? in any case, wouldn't it be an overall genome statistics in the direction of region insertion correlation (or anova) accross replicates vs accross variants.

##minor comments

Fig. 1D DNA footprinting figure is very hard to see. Maybe negative and positive samples need to be displayed separately.

Fig. 7E legend might have a coloring issue. Colors do not map to the barchart.

Version 1:

Reviewer comments:

Reviewer #1

(Remarks to the Author)

The authors have addressed my concerns.

Reviewer #2

(Remarks to the Author)

The authors have fully satisfied all my concerns.

Reviewer #3

(Remarks to the Author)

no further comments from my side

We are grateful to the reviewers for their careful reading of our manuscript, and their helpful suggestions and comments. Point-by-point responses (in red) to the reviewers' comments are below:

Reviewer #1:

This manuscript from Dyda probes the transposon end binding characteristics of the piggyBat transposon, a homolog of the widely used piggybac transposon. Very detailed mechanistic studies of the binding sites and their affinities are accompanied by a partial cryo-EM structure of the transposase bound to a single end. Like piggybac, the piggybat transposase's activity is stimulated by removal of the phosphorylation site in the N-terminus. Perhaps the most interesting finding is that additional internal transposase binding sites in the piggyBat ends appear to be inhibitory to piggyBat's function.

Though the manuscript claims that the piggyBat transposon is a promising alternative to the piggyBac transposon, it isn't clear from the findings in the manuscript what is so attractive about this system compared to piggyBac. At best it appears to be a homolog that functions similarly in primary human cells. Therefore, aside from the interesting finding that internal binding sites inhibit transposition I wonder: what is the broader significance of the study aside from this? It would benefit the authors and the Nature Communications audience to have this clearly articulated.

We appreciate the reviewer's point that the results presented in this manuscript do not demonstrate that *piggyBat* is superior to *piggyBac*. Our goal, as now stated more explicitly in the 3rd paragraph of the introduction, was to begin the exploration of the system to determine if it may have properties that are complementary or better than those of *piggyBac*. Part of that exploration was our effort to increase its overall transposition level to be similar to that of *piggyBac*. We believe that our work so far on *piggyBat* has demonstrated that, with modifications, its overall transposition activity can be increased to the level comparable to that of the hyperactive *piggyBac* versions. This means that *piggyBat* could be an addition to the genome modification toolkit of *piggyBac*-like transposons. Future studies will tell whether *piggyBat* also displays genome-wide integration profiles that have been a limitation of *piggyBac* applications due to safety concerns. Furthermore, it will be interesting to see whether *piggyBat* is more amenable for engineering applications aimed at specific targeting that so far have had only limited success with *piggyBac*.

Our discovery that internal binding sites inhibit transposition is part of a broader message that one perhaps should be cautious about classifying transposable elements and their potential for genome engineering based on current levels of transposition activity (see, for example, Cell 187, 3741 (2024); July 11th). There is an entire alternate dimension to the discovery and subsequent exploitation of transposable elements that can benefit from the knowledge of factors that control transposition activity. This has been difficult in the past where the main tool was random mutagenesis of the transposase itself. We have added a paragraph to this effect to the Discussion section:

"Our results are also a caution that, although current transposition activity is one valuable metric for assessing the genome engineering potential of newly-discovered DNA transposons, a better understanding of other features such as the arrangement of repeated motifs within transposon ends and the consequence of their presence can be a fruitful avenue of exploration. Future studies will tell whether *piggyBat* displays genome-wide integration profiles that have been a limitation of *piggyBac* applications due to safety concerns. Furthermore, it will be interesting to see whether *piggyBat* is more amenable for engineering applications aimed at specific targeting that so far have had only limited success with *piggyBac*."

Reviewer #2:

The activation of a mammalian DNA transposase and the process used to achieve it presented in this manuscript are impressive on two fronts: (1) it is likely to provide a useful new genetic tool and (2) it demonstrates the power of biochemical and structural data combined with insightful thinking when trying to optimize tools – the activation reported here would likely not have been possible even with the most powerful modern selection methods.

The work that I am most qualified to judge - the biochemical and structural sections - appears to be carefully and rigorously carried out.

Specific comments:

1) (very minor issue) Lines 77-80: there an abrupt transition from the idea that half the human genome derives from transposons to the next sentence that talks about DNA transposons, which (as far as we know) aren't active in the human genome.

We see what the reviewer means, and have tried to solve the problem by breaking the text into a new paragraph at that point.

2) Line 119: I see 4bp, not 15bp TIRs in Figure 1. This is confusing.

The reviewer is correct that this was confusing. It was originally reported in Ref 11 (Ray et al., 2008) that *piggyBat* has 15 bp TIRs. We have now added the reference and underlined the first 15 bp on the transposon ends in Figure 2 (we think this is the figure to which the reviewer is referring).

3) Line 132: how was "saturated" determined?

Saturated means that no further protection was observed at higher protein concentrations (as opposed to the visible difference between 340 and 700 nM protein). The next higher concentration used was 900 nM. We have added this explanation to the text.

4) Line 166: "transposon ends are typically organized as inverted repeats" is confusing. Do you mean the TIRs (one sequence repeated once on each end)? Or the same sequence twice on

each end, in inverted orientation (a palindrome)? I can only think of piggyBac as an example of the latter. In general, the discussion of motifs under the sub-heading “identification of DNA sequence motifs ...” is long-winded and confusing. The figure is nice, so a more concise summary in the text might be better.

As suggested, we have shortened this section and have removed the statement about inverted repeats at transposon ends, since there are inverted repeats (palindromes) within inverted repeats (TIRs) and it was indeed unnecessarily confusing.

5) Figure 1 – the peaks in the footprinting traces are so sharp that it is quite hard to tell red or blue from black in the traces. Can you show them offset vertically? Subtract the two curves for the main figure and put the raw curves in the supplement?

Both Reviewer #2 and Reviewer #3 comment on the depiction of the footprinting traces. We are afraid that offsetting them vertically in the main figure makes it quite hard to compare, and fully separating the DNA controls from the traces with protein (as per Reviewer #3 suggestion) would make the figure twice as large or each trace half the size (worsening the situation). We propose that the raw curves included in the Source Data file allows the interested reader to customize the display of the footprinting traces as they see fit, to either offset them or separate them. We have also now included all of the separated traces as Supplementary Figure 1.

6) Figure 1 – how do you explain the footprint around -30 on the RE?

There are no obvious motifs around the -30 bp region on the RE to suggest specific binding, so it might be an end-binding effect since it was consistently observed. However, we did not observe the same effect on the LE so we really do not have a good explanation. Although we are not enthusiastic about doing so, we have added the following to the text:

"As this protection was consistently observed, it might be related to the binding of a DNA end."

7) Having looked at the map myself in Coot, I think figure S1F slanders it. The figure made me doubt that the DNA sequence register could be established, or that individual bases could be seen rather than a continuous stack. Looking at the map reassured me that the section on DNA sequence recognition is not over-interpreted as I'd feared. I suggest adding close-up pictures of the density supporting the protein – DNA interactions, such as those two R's diving into the major groove.

As suggested by the reviewer, we have replaced this figure (now Supplementary Figure 3f) with a close-up picture of the density about the two arginines.

8) R497 might be contributing affinity to the protein-DNA interaction, but I don't believe it can contribute specificity because all 4 bases have H-bond acceptors at the same spot in the minor groove.

The reviewer is correct, and we have removed the word "specificity".

9) Can you say more about HOW the protein induces curvature in the DNA? Is it a matter of overall surface complementarity to the dimer? Asymmetric neutralization of one side of the

DNA? I also noted minor groove compression in the center, where the binding site is A/T (A/T richness tends to favor narrow minor grooves – see ancient work on 434 repressor).

The reviewer is correct that the shape of the dimer and the (relatively) large monomer-monomer and monomer-DNA interfaces pull the DNA towards the dimer, causing a large bend. A significant difference between piggyBat and piggyBac complexes is that, in the piggyBat case, there are protein/protein interactions between the CRD closest to the core domains and the core domains themselves. The result is that the CRD dimer sits snugly on the top of the cores. This eliminates the configurational freedom relative to the core that the piggyBac CRD dimer has. The end result is that a fixed DNA binding surface is created in the piggyBat case that is compatible with the bent DNA. This was also reflected in the much better quality potential density we observed for the piggyBat CRD dimer when compared to that of piggyBac. The central A/T richness (present in all three 44bp repeats and also in the center of the piggyBac palindrome) may well also aid bending. We have added the following to the text where the bend is described:

"and is aided by shape complementarity between the dimer and DNA as well as the A/T richness in the center of the palindrome. The transposase dimer DNA binding surface is compatible with the bent DNA and this surface is stable due to the fixed three-dimensional relationship between the core domains and the CRD closest to the core, supported by protein/protein interactions between them."

10) Line 369 – “figure 6D” – did you mean 7e?

We thank the reviewer for catching that error, and we meant current Figure 7d.

11) Line 451 – “the major restriction” – because it is hard to know how many additional unknown mechanisms are tuning down transposition in vivo, this should be changed to “a major restriction”.

The reviewer is of course correct. The change has been made.

12) Line 506 – I see no synaptic complexes at all in figure 3C?

The reviewer is correct and we have removed the word "synaptic"; we have corrected this to simply "multiple complexes".

In fact, I suspect that different fluorophores were used on the two transposon ends in order to detect synaptic complexes, and that no EMSA lanes are shown with both DNAs in the same tube because that part of the experiment failed? I would be in favor of just showing the failed experiment if that is the case – it would explain why so much effort was put into cryoEM of the single-end complex (which was, in fact, informative for protein engineering).

Yes, a great deal of ultimately unsuccessful effort was put into detecting synaptic complexes by EMSA. We now state this explicitly in the text and have added Supplementary Figure 2 to the supplemental material:

"We also attempted to detect synaptic complexes by EMSA in which pBat bound a LE and RE oligonucleotide simultaneously. However, we were ultimately unsuccessful, despite varying the binding conditions and order of addition (two examples are shown in Supplementary Figure 2)."

It may or may not be appropriate to add to this paper, but out of sheer curiosity, I would love to

know how well (or poorly) the new alphafold3 server correctly docks the CRD fold onto dsDNA, since structures of that fold bound to DNA couldn't have been in its training set.

This is an excellent suggestion, and as we also were wondering about the outcome of such an in-silico experiment, we performed it using the AlphaFold 3 (AF3) server. We asked AF3 to predict the interaction between the DNA sequence we used in the Cryo-EM work and the piggyBat CRDs, assuming that it interacts with the DNA as a dimer. The result is shown in the image below:

On the left is the AF3 result. On the right is the cryo-EM structure, with everything but the DNA and the CRD dimer removed. The palindromic DNA sequence that is the CRD dimer recognition motif is colored in magenta. The DNA of the AF3 prediction is longer as we have included all those nucleotides that were present in the cryo-EM experiment, some of which were not visible in the potential density map.

There are several notable features of the AF3 model: AF3 correctly identified the fold of the CRD and also correctly placed the two binding loops in the major grooves. Furthermore, AF3 essentially correctly identified the binding sequence where the two CRDs are bound.

However, the orientation of the CRD domains relative to the DNA is very different in the AF3 prediction relative to the cryo-EM structure. Both domains are rotated by about 180° about an axis that is approximately the pseudo two-fold axis of dsDNA perpendicular to the double helix and in this case located in the center of the major groove where the CRD binding loops bind. These rotations create a CRD dimer interface that is inconsistent with the cryo-EM structure and involves the upstream terminal of the CRDs that in the cryo-EM structure are involved in DNA binding in the adjacent minor grooves.

We conclude that while the AF3 model has features consistent with experimental reality, the actual AF3 predicted binding mode is not correct. We prefer not to include these results in the manuscript, given the results may not be trivially reproducible once the Protein Data Bank is updated to include our cryo-EM structure.

Reviewer #3:

Hickman et al report interesting re-engineering of the piggybat transposon. They report a novel inhibitory mechanism and they achieve remarkable activity enhancement. The authors leverage previous work from a more established piggybac system.

##general comments

-Figure 7 provides data on CRD domain. Page 13 or during the discussion (page 18) would benefit for a wider discussion on the structural insights explaining differences in V1 vs V2

We apologize for the vague discussion of the differences between V1 and V2 in the original submission. We now include the following on (original) page 13:

"Specifically, we had not anticipated the ~43 Å distance between the C-terminus of the first CRD and the first ordered residue (S494) of the second CRD. From the cryo-EM structure of the pre-synaptic complex, the point of closest approach between the C-terminus of the first CRD and the second monomer was at residue 479, so we therefore generated pBat-2xCRDV2 comprising residues 1-572+480-572 (**Figure 7g**)."

-I would be great to provide more clarity on the observations from Fig. 8

1) are the differences observed in B significant?

We thank the reviewer for the comment and have clarified our observations in the text where we discuss Fig. 8b. In particular, our analysis of genomic insertion categories (intronic + intergenic, and coding sequence (CDS) + untranslated regions (UTR)) did not reveal any significant differences between wild type (WT) and 4StoA-2xCRD pBat insertions (Chi-square test, $p > 0.05$). However, as the reviewer noted, we do observe a lower insertion frequency at intergenic regions of both constructs compared to the expected genomic representation and the genome-wide distribution of TTAA motifs that we computed based on the current genome release (GCA_000001405.15) for reference. This lower-than-expected representation of intergenic regions across all experimental conditions might be due to the reduced potential for insertions into constitutive heterochromatin, the increased fraction of repeat regions (resulting in reduced read mappability), and the larger proportion of single nucleotide polymorphisms and other sequence variants in intergenic regions compared to the reference genome. While all these factors could contribute to the lower representation of intergenic regions, we did not observe any differences between the WT and 4StoA-2xCRD pBat.

2) it results unclear to me what is being reported in D. 3 replicated of the two different variants? in any case, wouldn't it be an overall genome statistics in the direction of region insertion correlation (or anova) accross replicates vs accross variants.

The reviewer is correct that these are three replicates using the two different variants. To better compare their insertion profiles, we are currently optimizing our genomic insertion sequencing protocol to obtain more saturated datasets. Given the overall low frequency of transposition events, especially for WT pBat, and the large size of the human genome, we observe only about 6% (average) overlap of individual insertions across different replicates. Therefore, we have focused our analyses on comparing broad annotation categories, genomic distribution across different chromosomes, and local sequence context between WT and 4StoA-2xCRD pBat.

Figure 8d was provided to illustrate a representative shape of transposition hotspots. We are open to moving this figure to the supplementary material or removing it altogether, depending on the reviewers' preference.

Our future efforts will focus on determining saturation curves and developing methods to obtain saturated datasets. This will allow us to compare the frequency of individual insertions and potentially integrate additional genomic and epigenetic features.

##minor comments

Fig. 1D DNA footprinting figure is very hard to see. Maybe negative and positive samples need to be displayed separately.

This was also brought up by Reviewer #2 and has been addressed in the previous section. The separate traces have now been presented in the supplemental material (Supplementary Figure 1).

Fig. 7E legend might have a coloring issue. Colors do not map to the barchart.

We thank the reviewer for catching this. We have now correctly colored the right-most bar, and its RGB values match those in the legend.